



# 1 Impact of HO₂ aerosol uptake on radical levels and O₃

# 2 production during summertime in Beijing

Joanna E. Dyson[1], Lisa K. Whalley[1,2]*, Eloise J. Slater[1,a], Robert Woodward-Massey[1],
Chunxiang Ye[3], James D. Lee[4,5], Freya Squires[4,b], James R. Hopkins[4,5], Rachel E. Dunmore[4],
Marvin Shaw[4,5], Jacqueline F. Hamilton[4], Alastair C. Lewis[4,5], Stephen D. Worrall[6], Asan
Bacak[7], Archit Mehra[8,c], Thomas J. Bannan[8], Hugh Coe[8,9], Carl J. Percival[10], Bin Ouyang[11], C.
Nicholas Hewitt[11], Roderic L. Jones[12], Leigh R. Crilley[13], Louisa J. Kramer[14], W. Joe. F.
Acton[14], William J. Bloss[14], Supattarachai Saksakulkrai[14], Jingsha Xu[14,d], Zongbo Shi[14], Roy
M. Harrison[14,e], Simone Kotthaus[15,16], Sue Grimmond[15], Yele Sun[17], Weiqi Xu[17], Siyao
Yue[17,18,19], Lianfang Wei[17,19], Pingqing Fu[17,18], Xinming Wang[20], Stephen R. Arnold[21],
Dwayne E. Heard[1]*
*[1] School of Chemistry, University of Leeds, LS2 9JT, UK.*
*[2] National Centre of Atmospheric Science, University of Leeds, LS2 9JT, UK.*
*[3] College of Environmental Sciences and Engineering, Peking University, Beijing, 100871,*
*China.*
*[4] Wolfson Atmospheric Chemistry Laboratories, Department of Chemistry, University of*
*York, Heslington, York, YO10 5DD, UK.*
*[5] National Centre of Atmospheric Science, University of York, Heslington, York, YO19 5DD,*
*UK.*
*[6] Aston Institute of Materials Research, School of Engineering and Applied Science, Aston*
*University, Birmingham, B4 7ET, UK.*
*[7] Turkish Accelerator and Radiation Laboratory, Ankara University Institute of Accelerator*
*Technologies, Atmospheric and Environmental Chemistry Laboratory, Gölbaşi Campus,*
*Ankara, Turkey.*
*[8] Centre of Atmospheric Sciences, School of Earth and Environmental Sciences, University*
*of Manchester, Manchester, M13 9PL, UK.*
*[9] National Centre for Atmospheric Sciences, University of Manchester, Manchester, M13*
*9PL, UK.*



*[10] Jet Propulsion Laboratory, California Institute of Technology, Pasadena, CA, USA.*
*[11] Lancaster Environment Centre, Lancaster University, Lancaster, LA1 4YW, UK.*
*[12] Department of Chemistry, University of Cambridge, Cambridge, UK.*
*[13] Department of Chemistry, York University, Toronto, ON, M3J 1P3, Canada.*
*[14] School of Geography, Earth and Environmental Sciences, University of Birmingham,*
*Birmingham, B15 2TT, UK.*
*[15] Department of Meteorology, University of Reading, Reading, UK.*
*[16] Institut Pierre Simon Laplace, École Polytechnique, Palaiseau, France.*
*[17] State Key Laboratory of Atmospheric Boundary Layer Physics and Atmospheric*
*Chemistry, Institute for Atmospheric Physics, Chinese Academy of Sciences, Beijing 100029,*
*China.*
*[18] Institute of Surface-Earth System Science, School of Earth System Science, Tianjin*
*University, Tianjin 300072, China.*
*[19] Minerva Research Group, Max Planck Institute for Chemistry, 55128 Mainz, Germany.*
*[20] State Key Laboratory of Organic Geochemistry, Guangzhou Institute of Geochemistry,*
*Chinese Academy of Sciences, Guangzhou, 510640, China.*
*[21] School of Earth and Environment, University of Leeds, LS2 9JT, UK.*
*[a]now at: The Hut Group, Unit 1 Icon Manchester, Manchester Airport, WA15 0AF, UK.*
*[b]now at: British Antarctic Survey, Cambridge, CB3 0ET, UK.*
*[c]now at: Chaucer, Part of Bip Group, 10 Lower Thames Street, London, EC3R 6EN[d]now at:*
*Beijing Hanzhou Innovation Institute Yuhang, Xixi Octagon City, Yuhang District, Hangzhou*
*310023, China.*
*[e]also at: Department of Environmental Sciences, Faculty of Meteorology, Environment and*
*Arid Land Agriculture, King Abdulaziz University, Jeddah, Saudi Arabia.*
*\*Correspondence to: D.E.Heard@leeds.ac.uk, L.K.Whalley@leeds.ac.uk*






**Abstract** The impact of heterogeneous uptake of $HO_2$ onto aerosol surfaces on radical
concentrations and the $O_3$ production regime in Beijing summertime was investigated. The
uptake coefficient of $HO_2$ onto aerosol surfaces, $\gamma_{HO_2}$, was calculated for the AIRPRO
campaign in Beijing, Summer 2017, as a function of measured aerosol soluble copper
concentration, $[Cu^{2+}]_{eff}$, aerosol liquid water content, $[ALWC]$, and particulate matter
concentration, $[PM]$. An average $\gamma_{HO_2}$ across the entire campaign of $0.070 \pm 0.035$ was
calculated, with values ranging from 0.002 to 0.15, and found to be significantly lower than
the value of $\gamma_{HO_2} = 0.2$, commonly used in modelling studies. Using the calculated $\gamma_{HO_2}$ values
for the Summer AIRPRO campaign, OH, $HO_2$ and $RO_2$ radical concentrations were modelled
using a box-model incorporating the Master Chemical Mechanism (v3.3.1), with and without
the addition of $\gamma_{HO_2}$, and compared to the measured radical concentrations. Rate of destruction
analysis showed the dominant $HO_2$ loss pathway to be $HO_2 + NO$ for all NO concentrations
across the Summer Beijing campaign with $HO_2$ uptake contributing < 0.3 % to the total loss of
$HO_2$ on average. This result for Beijing summertime would suggest that under most conditions
encountered, $HO_2$ uptake onto aerosol surfaces is not important to consider when investigating
increasing $O_3$ production with decreasing $[PM]$ across the North China Plain. At low $[NO]$,
however, i.e. < 0.1 ppb, which was often encountered in the afternoons, up to 29% of modelled
$HO_2$ loss was due to $HO_2$ uptake on aerosols when calculated $\gamma_{HO_2}$ was included, even with the
much lower $\gamma_{HO_2}$ values compared to $\gamma_{HO_2} = 0.2$, a results which agrees with the aerosol-
inhibited $O_3$ regime recently proposed by Ivatt et al., 2022. As such it can be concluded that in
cleaner environments, away from polluted urban centres where $HO_2$ loss chemistry is not
dominated by NO but where aerosol surface area is high still, changes in PM concentration and
hence aerosol surface area could still have a significant effect on both overall $HO_2$
concentration and the $O_3$ production regime.
Using modelled radical concentrations, the absolute $O_3$ sensitivity to $NO_x$ and VOC showed
that, on average across the summer AIRPRO campaign, the $O_3$ production regime remained
VOC-limited, with the exception of a few days in the afternoon when the NO mixing ratio
dropped low enough for the $O_3$ regime to shift towards $NO_x$-limited. The $O_3$ sensitivity to VOC,
the dominant regime during the summer AIRPRO campaign, was observed to decrease and
shift towards a $NO_x$ sensitive regime both when NO mixing ratio decreased and with the
addition of aerosol uptake. This suggests that if $[NO_x]$ continues to decrease in the future, ozone
reduction policies focussing solely on $NO_x$ reductions may not be as efficient as expected if

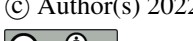



[PM] and, hence, HO$_2$ uptake to aerosol surfaces, continues to decrease. The addition of aerosol
uptake into the model, for both the $\gamma_{HO_2}$ calculated from measured data and when using a fixed
value of $\gamma_{HO_2} = 0.2$, did not have a significant effect on the overall O$_3$ production regime across
the campaign. While not important for this campaign, aerosol uptake could be important for
areas of lower NO concentration that are already in a NO$_x$-sensitive regime.

# 1   Introduction

Climate change and air quality are two significant environmental issues faced by society today
with the drive to net zero emissions by 2050 becoming increasingly important to remain
consistent with the long-term anthropogenic temperature warming outcome of below 1.5 °C as
set out by the Paris Agreement in 2016. Increasing anthropogenic emissions have caused not
only an increase in atmospheric warming, but also a deterioration in atmospheric air quality: a
concern due to both short and long term negative health effects seen as a product of poor air
quality such as respiratory and cardiovascular diseases and cancer (Brauer et al., 2016; Gakidou
et al., 2017), in addition to a variety of negative effects on the environment such as increased
soil acidification and the ensuing damage to vegetation and crop yield as a by-product of
increasing acidity of rain (Forster et al., 2007).
Ambient air pollution has become a serious issue globally, specifically in large urban areas
effected by anthropogenic emission sources. Due to rapid industrialisation, Chinese megacities
in particular face significant environmental and health challenges from the decline in air quality
following urbanisation, with areas such as the Beijing-Tianjin-Hebei area in the North China
Plain (NCP) suffering from seasonal extreme pollution episodes as a consequence (Wang,
2021; Jin et al., 2016).  In terms of human health, the most important pollutants in many regions
are ground level O$_3$, NO$_x$ (NO$_2$ and NO) and particulate matter. Nitrogen dioxide (NO$_2$) can be
directly emitted into the atmosphere from high temperature combustion sources or can be
formed via the reaction of nitrogen monoxide (NO) with an oxidising species in the
troposphere, such as HO$_2$, leading to the formation of hydroxyl radical (OH) (Ye et al., 2017).
Ozone, while vital in the stratosphere to protect the earth from harmful UV radiation and
excessive planetary heating, is toxic to both plant and human life at ground level and can react
with NO to form NO$_2$. Particulate matter is emitted anthropogenically and biogenically and can
play a role in the warming and cooling of the atmosphere due to the ability of aerosols to absorb
or scatter IR radiation depending on their composition. High levels of particulate matter, NO$_x$





and tropospheric $O_3$ in areas of low atmospheric mixing lead to photochemical smog and the
reduction of visibility characteristic of extreme pollution episodes.
The concentration of pollutants and trace gases in the troposphere is controlled not only by
emission levels but also by the oxidation capacity of the atmosphere which is determined
largely by the concentration of the hydroxyl radical (OH) and the closely coupled hydroperoxyl
($HO_2$) radical, referred to collectively as $HO_x$ radicals. Known for their role in chemical
oxidation processes in the atmosphere, OH and $HO_2$ are vital species when considering climate
change and air pollution. The OH radical is the main daytime tropospheric oxidant, with a
major role as a source of ground level ozone ($O_3$) (Levy, 1971) and as a sink for both
atmospheric pollutants, such as methane, and other radical species. The OH radical also has a
role in the formation of secondary pollutants including secondary organic aerosols (SOAs)
formed via the oxidation of volatile organic compounds (VOCs). OH and $HO_2$ radicals are
closely linked, due to the recycling of $HO_2$ to give OH, either via the reaction with NO or CO,
with the dominant loss pathway of $HO_2$ in polluted regions being the reaction with NO to form
OH (for example, as shown in Beijing by Slater et al., 2020; Whalley et al., 2021). As such,
understanding the sources and sinks of both OH and $HO_2$ within the troposphere is crucial to
fully understand the concentration and distribution of trace atmospheric species associated with
climate change and poor air quality.
Observed $HO_2$ concentrations from field measurements frequently can-not be fully explained
by atmospheric chemistry models which often have a tendency to over-predict $HO_2$ in low $NO_x$
conditions (Kanaya et al., 2007; Commane et al., 2010; Whalley et al., 2010; Whalley et al.,
2021; Slater et al., 2020; Sommariva et al., 2004). Following the ClearfLo campaign in London
2012, zero-dimensional modelling showed an over-prediction of $HO_2$ by up to a factor of 10 at
low $NO_x$ which was attributed to uncertainties in the degradation mechanism of complex
biogenic and diesel-related VOC species at low $NO_x$ (Whalley et al., 2018). Over-prediction of
$HO_2$ is also commonly thought to be due, in part, to lack of understanding of $HO_2$ uptake onto
aerosol surfaces. A 2014 modelling study by Xue et al., 2014 focussing on the transport,
heterogeneous chemistry and precursors of ground level ozone in Beijing, Shanghai,
Guangzhou and Lanzhou, identified $HO_2$ uptake as a source of uncertainty when considering
ozone production, with uptake onto aerosols having the largest effect on $HO_2$ concentration in
Beijing where aerosol loadings were the highest.





While the impact of $HO_2$ uptake on $HO_x$ concentrations has been calculated to vary from ~10-
40 % (Jacob, 2000; Whalley et al., 2010; Whalley et al., 2021; Slater et al., 2020; Mao et al.,
2010; Li et al., 2019; Li et al., 2018) globally, often a single value of $\gamma_{HO_2}= 0.2$ is used within
models, as recommended by Jacob, 2000. Previous experimental studies report uptake
coefficients which span several orders of magnitude, however, and vary largely based on the
state of the aerosol and whether transition metal ion catalysis is involved.  For dry inorganic
salt aerosols values as low as $\gamma_{HO_2} < 0.002$ have been reported (Cooper and Abbatt, 1996;
Taketani et al., 2008; George et al., 2013) increasing to up to $\gamma_{HO_2}= 0.2$ for aqueous aerosols
(Thornton and Abbatt, 2005; Taketani et al., 2008; George et al., 2013). Previous experimental
studies report much higher $\gamma_{HO_2}> 0.4$ for Cu-dopped aqueous aerosols (Thornton and Abbatt,
2005; Mozurkewich et al., 1987; Taketani et al., 2008; George et al., 2013; Lakey et al., 2016).
Recently, larger values of $\gamma_{HO_2}$ have been measured experimentally from samples taken offline
at Mt. Tai (0.13-0.34) and Mt. Mang (0.09-0.40) in China by Taketani et al., 2012, while
another study in Kyoto, Japan, directly measured $\gamma_{HO_2}$ values under ambient conditions from
0.08 to 0.36 (Zhou et al., 2020). With $\gamma_{HO_2} > 0.1$, $HO_2$ concentrations can be significantly
influenced particularly in areas of low [NO] and/or high aerosol loadings (Lakey et al., 2015;
Matthews et al., 2014; Mao et al., 2013; Zhou et al., 2021; Martinez et al., 2003).
Following multiple policies implemented across China in response to the poor air quality
"crisis", a number of studies have reported a decrease in $NO_x$ and $PM_{2.5}$ emissions in China
(Jin et al., 2016). Liu et al., 2017 reported $NO_x$ ($NO_2$ + NO) emissions over 48 Chinese cities
to have decreased by 21 % in the period of 2011-2015, supported by observed declines in $NO_x$
emissions reported by other studies (Krotkov et al., 2016; Liu et al., 2016; Miyazaki et al.,
2017; Van Der A et al., 2017). Ma et al., 2016b reported a mean annual decrease in $PM_{2.5}$ of
0.46 µg m$^{-3}$ between 2008-2013, while Lin et al., 2018 reported an average decrease of 0.65
µg m$^{-3}$ yr$^{-1}$ between 2006-2010 increasing to a decline of 2.33 µg m$^{-3}$ yr$^{-1}$ for the period of
2011-2015. In contrast to the observed decrease in $NO_x$ and $PM_{2.5}$ emissions, several studies
have reported increasing $O_3$ levels. Ma et al., 2016a reported a maximum daily average 8h
mean (MDA8) increase in $O_3$ concentrations of 1.13 ppb yr$^{-1}$ for the period between 2003-2015
at a rural site north of Beijing while satellite observations suggested ground level ozone had
increased ~7% for the period between 2005-2010 (Verstraeten et al., 2015). A recent study by
Silver et al., 2018 also observed a significant increase in $O_3$ concentrations with median MDA8
increasing at a rate of 4.6 µg m$^{-3}$ yr$^{-1}$ across China.




A 2018 modelling study using the regional model GEOS-Chem by Li et al., 2018 suggested
the increase in $O_3$ across China between 2013-2017 could be attributed to the decrease in $PM_{2.5}$,
with changes in $PM_{2.5}$ being a more important driver of increasing $O_3$ trends than $NO_x$ and
VOC emissions for the period studied. It was proposed that a decrease in $PM_{2.5}$ emissions had
led to a decrease in loss of $HO_2$ via aerosol uptake resulting in an increase in $HO_2$ concentration,
and a proportional increase in the loss of $HO_2$ via NO leading to $NO_2$ which, when photolyzed,
forms $O_3$ leading to an increase in $O_3$ (Li et al., 2018). However, analysis of measured radical
budget from a field campaign in the North China Plain in Summer 2014 with a calculated $\gamma_{HO_2}$
of $0.08 \pm 0.13$, showed no evidence for a significant impact of $HO_2$ heterogeneous chemistry
on radical concentrations in North China Plain, concluding that reduced $HO_2$ uptake was
unlikely to therefore be the cause of increasing $O_3$ levels in the North China Plain (Tan et al.,
2020). Using a novel parameterisation developed by Song et al., 2020 in the framework of the
resistor model to take into account the influence of aerosol soluble copper, aerosol liquid water
content and particulate matter concentration on $HO_2$ uptake, and the Multiphase Chemical
Kinetic box model (PKU-MARK) to assess the impact of $HO_2$ uptake on the $O_3$ budget for
Wangdu Campaign in 2014, Song et al., 2022 concluded that $HO_2$ heterogeneous processes
could decrease the $O_3$ production rates by up to 6 ppbv hr$^{-1}$, particularly in the morning VOC-
limited regime.
In this study, the new parameterisation introduced by Song et al., 2021, hereafter referred to
solely as the Song parameterisation, coupled with measured data from the Summer AIRPRO
campaign in Beijing 2017 was used to calculate a time series of the $HO_2$ uptake coefficient,
which was then used to investigate the impact of heterogeneous uptake of $HO_2$ onto aerosol
surfaces on the $HO_2$ radical budget in Summertime Beijing using the Master Chemical
Mechanism and the impact on the $O_3$ regime. We will test the hypothesis that reduced $HO_2$
uptake due to a reduction in $PM_{2.5}$ concentration is a significant driver of the recent increase in
ozone concentrations in China.
## 2   Experimental
### 2.1   Campaign overview and site description
As part of the Atmospheric Pollution and Human Health (APHH) in a Chinese Megacity
programme, the University of Leeds took simultaneous measurements of OH, $HO_2$, $RO_2$ and
OH reactivity ($k_{OH}$), in addition to measurements of HCHO and photolysis rates, during two





field campaigns at an urban site in Winter 2016 and Summer 2017 in Beijing, with the aim to
study the chemical and physical processes governing gas and particle pollution and
meteorological dynamics in the Beijing region and the links between the two (Shi et al., 2019;
Slater et al., 2020; Whalley et al., 2021). The two field campaigns in Beijing were part of the
AIRPRO (The integrated study of AIR pollution PROcesses in Beijing) project within the
APHH programme, described fully by Shi et al., 2019.
For the summer AIRPRO campaign, the official science period was from 23$^{rd}$ May 2017 to the
22$^{nd}$ June 2017, with observations taking place at the Institute of Atmospheric Physics (IAP)
within the Chinese Academy of Sciences, located between the third and fourth ring roads in
central Beijing within 100 m of a major road, making local traffic emission sources an
important source of pollution during measurement period. All instrumentation for the campaign
was located at this site, housed within nine shipping containers surrounding a meteorological
tower. Further details of the instrumentation and measurement site can be found in Shi et al.,
226   2019.

**2.2   FAGE instrumentation description**
The University of Leeds Fluorescence Assay by Gas Expansion (FAGE) instrument made
measurements of OH, HO$_2$ and RO$_2$ radicals and OH reactivity ($k_{OH}$). The FAGE instrument
set up is described fully in Whalley et al., 2018 while the OH reactivity instrument set up is
described fully in Whalley et al., 2016. Both instruments are also described fully in Slater et
al., 2020 and so only a brief description is given here.
Two cells, a HO$_x$ cell and a RO$_x$ cell connected together with a side arm, were used to take
radical measurements from the roof of the Leeds FAGE lab container. A RO$_x$LIF flow reactor
was also coupled to the RO$_x$ cell to allow for detection of RO$_2$ (total, complex and simple) as
described by Fuchs et al., 2008. The HO$_x$ cell took sequential measurements of OH and the
sum of OH and HO$_2$, by the addition of NO (Messer, 99.5 %), which titrated HO$_2$ to OH for
detection by Laser Induced Fluorescence (LIF) at 308 nm.
The RO$_x$LIF reactor operated in 2 modes: a 'HO$_x$ mode' where a flow of CO (10 % in N$_2$) was
added to ambient sampled air close to the pin hole to convert all ambient OH to HO$_2$; and a
'RO$_x$ mode' where NO (500 ppmv in N$_2$) was added in addition to the CO flow to convert all
RO$_2$ into OH before all OH was then rapidly converted by CO into HO$_2$. The air from the
RO$_x$LIF reactor was then drawn into the FAGE low pressure fluorescence cell, whereupon pure





NO (Messer, 99.5 %) was injected to convert $HO_2$ to OH. In $HO_x$ mode, the sum of OH, $HO_2$
and complex $RO_2$ was measured, while in $RO_x$ mode, the sum of OH, $HO_2$ and total $RO_2$ was
measured. From this the concentration of complex $RO_2$ and $HO_2$/OH from $RO_x$ can be
determined.
An Inlet Pre-Injector was used attached to the $HO_x$ cell to remove ambient OH by injecting
propane directly above the inlet of the cell. This leads to a background measurement while the
laser is still online to the OH transition; this background is known as $OH_{CHEM}$. $OH_{CHEM}$ includes
signal from laser scatter and scattered solar radiation and any fluorescence signal from any OH
generated inside the cell from an interference precursor. By comparing $OH_{CHEM}$ to the signal
generated when the 308 nm laser tuned off the OH transition, $OH_{WAVE}$, the contribution of any
interference can be identified. While the laser is offline, $OH_{WAVE}$, any signal seen is from laser
scattered light and scattered solar radiation. Agreement between $OH_{WAVE}$ and $OH_{CHEM}$ was
generally very good during the Summer AIRPRO campaign with an overall orthogonal
distance regression slope of $1.103 \pm 0.017$, with the exception of an interference seen when $O_3$
levels were elevated (see Woodward-Massey et al., 2020 for details).
## 2.3  Determination of aerosol soluble copper concentration through ICP-
## MS Analysis
The soluble copper ion concentration was determined by analysing the effluent extracted from
quartz filter samples taken daily for the entire campaign using Inductively Coupled Plasma
Mass Spectrometry (ICP-MS). A 6 $cm^2$ punch from each large quartz filter $PM_{2.5}$ sample was
cut and put in a 15 mL extraction tube and extracted with 10 mL ultrapure water (18.2 MΩ
cm) under ultrasonication for 60 minutes at below 35 ºC. The sample was then shaken by a
temperature-controlled shaker at 4 ºC for 3 hours at approximately 60 cycles $min^{-1}$. After
filtering through a filter syringe, 8 mL of effluent was transferred to a new 15 mL metal free
tube, and 2 mL of 10% $HNO_3$ was added to make a 10 mL 2% $HNO_3$ extract solution which
was then analysed to determine the soluble copper ion concentration using ICP-MS.
## 2.4  MCM v3.3.1 box model description
The Master Chemical Mechanism (MCM$v$3.3.1) is a near-explicit mechanism which describes
the gas-phase degradation of a series of primary emitted VOC's in the troposphere. The
mechanism considers the degradation of 143 VOC's and contains ~17000 elementary reactions
of 6700 species (Whalley et al., 2013).



The model was constrained to measurements of NO, NO$_2$, O$_3$, CO, HCHO, HNO$_3$, HONO,
PAN, H$_2$O vapour, temperature, pressure, $j$(O$^1$D), $j$(HONO), $j$(NO$_2$), $j$(ClNO$_2$), $j$(HOCl),
$j$(ClONO$_2$) and specific VOC species measured using GC-FID (gas chromatography with
flame ionisation) and PTR-ToF-MS (proton-transfer reaction time of flight mass
spectrometry). The measured species were input into the model at a time resolution of 15
minutes, with species measured at a higher time resolution averaged up to 15 minutes and those
measured at a lower time resolution interpolated to give a value every 15 minutes.  The full list
of all species constrained in the model is shown in Table 1.

| Type | Species |
|---|---|
| **Gas-phase inorganic species** | NO, NO$_2$, O$_3$, CO, HNO$_3$, HONO, H$_2$O, SO$_2$, ClNO$_2$, HOCl |
| **Gas-phase organic species** | HCHO, PAN, CH$_4$, C$_2$H$_6$, C$_2$H$_4$, C$_3$H$_8$, C$_3$H$_6$, isobutane, butane, C$_2$H$_2$, trans-but-2-ene, but-1ene, Isobutene, cis-but-2-ene, 2-Methylbutane, pentane, acetone, 1,3-butadiene, trans-2-pentene, cis-2- pentene, 2-methylpetane, 3-methypetane, hexane, isoprene, heptane, benzene, toluene, nonane, decane, undecane, dodecane, o-xylene, CH$_3$OH, CH$_3$OCH$_3$, 2-ethyltoluene, 3-ethyltoluene, 4-ethyltoluene, ethylbenzene, CH$_3$CHO, C$_2$H$_5$OH, α-pinene, limonene, isopropylbenzene, propylbenzene, m-xylene, p-xylene, 1,2,3-trimethylbenzene, 1,2,4-trimethylbenzene, 1,3,5-trimethylbenzene. |
| **Photolysis rates** | $j$(O$^1$D), $j$(HONO), $j$(NO$_2$), $j$(ClNO$_2$), $j$(HOCl), $j$(ClONO$_2$) |
| **Other** | Mixing height, aerosol surface area |

**Table 1.** Full description of measured species during Summer AIRPRO campaign constrained within the model
The different model scenarios referred to in this study are described in full below:

285        1. **MCM_base:** The base model run constrained to species described in Table 1.
286        2. **MCM_gamma:** The base model including heterogeneous HO$_2$ uptake onto
287           aerosols with $\gamma_{HO_2}$ calculated from parameterisation developed by Song et al., 2020.
288        3. **MCM_SA:** The base model including heterogeneous HO$_2$ uptake, this time with
289           $\gamma_{HO_2}$ fixed at 0.2, as commonly used within models and recommended by Jacob,
290           2000.





## 2.5 Description of the "Song parameterisation"


A large uncertainty in determining the effect of $HO_2$ uptake onto the surface of aerosol particles
is the lack of understanding of the dependence of $\gamma_{HO_2}$ on Cu (II)/transition metal ion
concentration within aerosols. Experimentally this dependence is quite well known from
laboratory studies (Mozurkewich et al., 1987; Thornton and Abbatt, 2005; George et al., 2013;
Mao et al., 2013), however the effective concentrations in ambient aerosols and the impact on
$\gamma_{HO_2}$ of aerosol liquid water concentration, [ALWC], has not been incorporated into models
before. A novel parameterisation was developed by Song et al., 2020 in the framework of the
resistor model to include the influence of aerosol soluble copper on the uptake of $HO_2$. The
new parameterisation for the uptake coefficient of $HO_2$ onto aerosols, as given in Song et al.,
2020, is as follows:

$$\frac{1}{\gamma_{HO_2}} = \frac{1}{\alpha_{HO_2}} + \frac{3 \times v_{HO_2}}{(4 \times 10^6) \times R_d H_{eff} RT \times \left(5.87 + 3.2ln\left(\frac{ALWC}{[PM] + 0.067}\right)\right) \times [PM]^{-0.2} \times [Cu^{2+}]_{eff}^{0.65}} \quad (1)$$

where $\gamma_{HO_2}$ is the uptake coefficient of $HO_2$ onto aerosols, $\alpha_{HO_2}$ is the mass accommodation
coefficient of $HO_2$, $v_{HO_2}$ is the mean molecular speed in cm s$^{-1}$, $R_d$ is the count median radius
of the aerosol in cm, $H_{eff}$ is the effective Henry's Law constant calculated from $H_{eff} =$
$H_{HO_2}\left(1 + \frac{K_{eq}}{[H+]}\right)$ where $H_{HO_2}$ is the physical Henry's Law constant for $HO_2$ (i.e. 3900
(Thornton et al., 2008)) in M atm$^{-1}$, $K_{eq}$ is the equilibrium constant for $HO_2$ dissociation (M),
and $[H^+]$ is the hydrogen ion concentration within the aerosol calculated from the pH (M), $R$
is the gas constant in cm$^3$ atm K$^{-1}$ mol$^{-1}$ (i.e. 82.05), $T$ is the temperature in K, $[ALWC]$ is the
aerosol liquid water content in µg m$^{-3}$ (which is related to the ambient relative humidity), $[PM]$
is the mass concentration of PM$_{2.5}$ in µg m$^{-3}$ and $[Cu^{2+}]_{eff}$ is the effective aerosol condensed-
phase soluble copper (II) ion concentration in mol L$^{-1}$.
The Song parameterisation can reportedly be used for urban environmental conditions of
aerosol mass concentrations between 10-300 µg m$^{-3}$; aqueous copper (II) concentrations of
$10^{-5}$–1 mol L$^{-1}$; and relative humidity between 40-90 %. However, for the Summer AIRPRO
campaign data, the minimum [ALWC] supported by the parameterisation was 14 µg m$^{-3}$, below
which the parameterisation returned negative values for $\gamma_{HO_2}$. As such, despite the average
calculated [ALWC] for the campaign being 6.9 ± 10 µg m$^{-3}$, a fixed value of 14 µg m$^{-3}$ was
used to calculate $\gamma_{HO_2}$ across the entire campaign.





## 3 Results and Discussion

### 3.1 Overview of field observations during summer AIRPRO campaign

Radical concentration measurements were taken throughout the official science period of the summer campaign, from 23/05/2017 to 22/06/2017, using the Fluorescence Assay by Gas Expansion technique. Alongside the radical observations and photolysis rate measurements made by the University of Leeds, there was a varied suite of supporting measurements operated by several universities and institutions. The supporting measurements used for the analysis and discussion in this study were provided chiefly by the Universities of York, Birmingham and Cambridge as detailed in Table 2.



| Instrument | Species measured | University | Reference |
|---|---|---|---|
| **FAGE** | OH, HO$_2$, RO$_2$ | Leeds | Whalley et al., 2010; Whalley et al., 2021; Slater et al., 2020 |
| **OH reactivity** | OH reactivity | Leeds | Stone et al., 2016; Whalley et al., 2021; Slater et al., 2020 |
| **Spectral Radiometer** | Photolysis rates | Leeds | Bohn et al., 2016 |
| **Filter Radiometer** | $j(O^1D)$ | Leeds | Whalley et al., 2010 |
| **Teledyne CAPS** | NO$_2$ | York | Smith et al., 2017 |
| **TEI 42c** | Total NO$_y$ | York | Smith et al., 2017 |
| **TEI 49i** | O$_3$ | York | Smith et al., 2017 |
| **Sensor box** | CO | York | Smith et al., 2017 |
| **DC-GC_FID** | C$_2$-C$_7$ VOCs and oVOCs | York | Hopkins et al., 2011 |
| **GCxGC-FID** | C$_6$-C$_{13}$ VOCs and oVOCs | York | Dunmore et al., 2015 |
| **BBCEAS** | HONO | Cambridge | Le Breton et al., 2014 |
| **TEI 42i** | NO | Birmingham | - |
| **LOPAP** | HONO | Birmingham | Crilley et al., 2016 |
| **SMPS** | Particle Size distribution | Birmingham | Wiedensohler et al., 2012 |
| **High volume sampler** | PM$_{2.5}$ filter samples, Aerosol copper | IAP | - |

**Table 2.** Measurements taken by universities and institutions during the Beijing Summer AIRPRO campaign.
These species are directly referred to in this chapter: full description of every instrument and measurement taken
can be found in Slater, 2020. IAP: Institute of Atmospheric Physics, Beijing. Time resolution of all instruments
was averaged up to or interpolated down to 15 minutes for modelling purposes with the exception of the PM$_{2.5}$
filter samples, of which there was only 1 sample taken a day.
The median average diurnals for important gas phase species (ppb) and $j(O^1D)$ (s$^{-1}$) measured
during the summer campaign are shown in Figure 1. $j(O^1D)$ showed a maximum at solar noon
peaking at $2.5 \times 10^{-5}$ s$^{-1}$. The diurnal variation in both NO and NO$_2$ was very distinct, with a
peak in NO at rush hour (~08:00) of ~ 8 ppb. NO decreased into the afternoon following this
morning peak to a minimum of 0.3 ppb. The low values of NO mixing ratio observed in the
afternoon were a result of high levels of O$_3$, peaking at 89 ppb at ~15:30, leading to increased
titration of NO + O$_3$ to give NO$_2$, the diurnal of which can be seen to peak in the morning at
~ 32 ppb at 06:30, coinciding with peak in traffic emissions. Conversely O$_3$ mixing ratio was
at a minimum of ~14 ppb during the morning traffic peak in NO. Due to the expected



accumulation of HONO overnight, HONO mixing ratio is highest in the morning, peaking
before 07:30 at ~ 7 ppb, after which HONO is lost rapidly via photolysis to give OH + NO.
This study will use these measured observations to compare modelled and measured
concentrations of OH, $HO_2$ and $RO_2$ radicals and investigate the effect of $HO_2$ uptake on radical
concentrations.

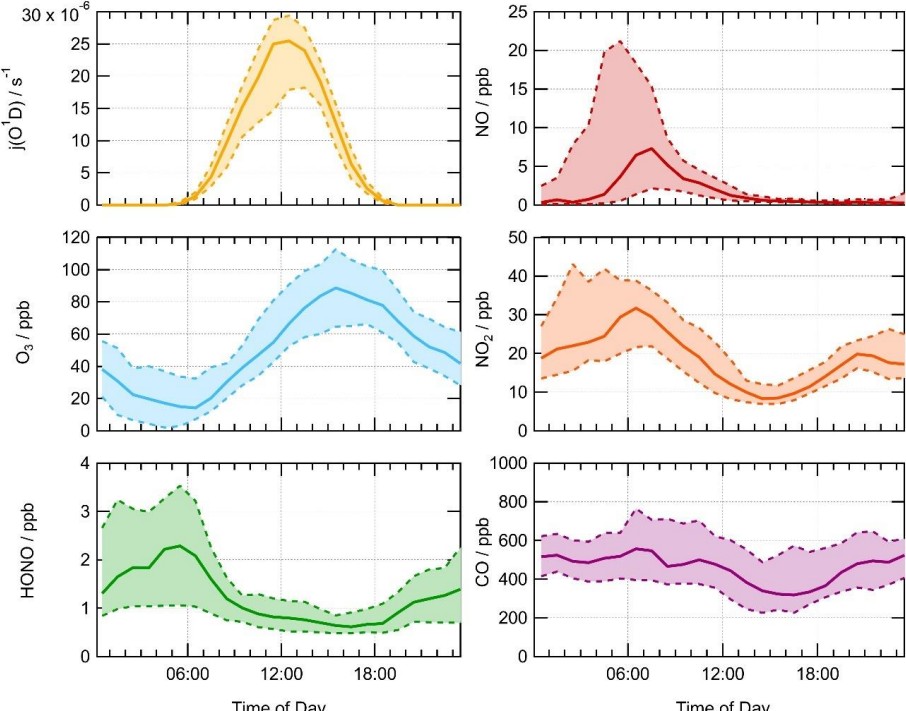

**Figure 1.** Average median diurnal profile for measured j(O¹D) (s⁻¹), $O_3$ (ppb), HONO (ppb), NO (ppb), $NO_2$ (ppb) and CO (ppb) for the Summer AIRPRO campaign. The dashed lines with shaded regions represent the 25th/75th percentiles. Diurnals show 60 minute averages, taken over the entire measurement period.

The majority of the Summer Beijing campaign occurred during a non-haze period, meaning
$PM_{2.5}$ concentrations remained below 75 µg m⁻³, only exceeding this on the 28/05, 31/05,
05/06, 07/06, 17/06 and 18/06/2017. The average median diurnal of $PM_{2.5}$ surface area
(cm² cm⁻³) is shown in Figure 2. $PM_{2.5}$ surface area concentration was available at a higher
resolution due to use of online particle sizers compared to filter samples taken daily to give
$PM_{2.5}$ mass concentration. $PM_{2.5}$ surface area was then averaged up to a time resolution of 15
minutes to be used in the model. No strong diurnal trend was seen, with an average across the
campaign of $5.5 \times 10^{-6}$ cm² cm⁻³, with a maximum surface area of $2.5 \times 10^{-5}$ cm² cm⁻³.



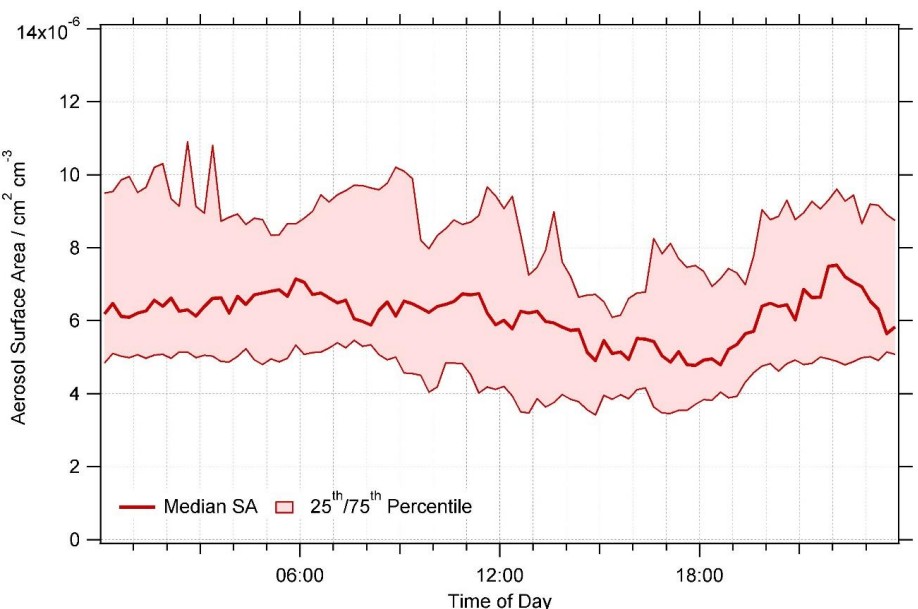

**Figure 2.** Average median diurnal of $PM_{2.5}$ aerosol surface area ($cm^2$ $cm^{-3}$) for Summer AIRPRO campaign. Data averaged up to 15 mins time resolution. The dashed lines with shaded regions represent the $25^{th}/75^{th}$ percentiles.

During haze periods in Beijing, it is expected that a strong correlation would exist between
$PM_{2.5}$ and $NO_x$, as seen in Winter Beijing AIRPRO campaign in 2016 (Slater et al., 2020).
However, during the Summer campaign, no strong correlation between $PM_{2.5}$ and $NO_x$ was
seen. The time series of NO (ppb) and $PM_{2.5}$ ($cm^2$ $cm^{-3}$) is shown in Figure 3. A correlation
plot of $PM_{2.5}$ aerosol surface area ($cm^2$ $cm^{-3}$) versus NO and $NO_2$ mixing ratio (ppb) is shown
in Figure 1 of Supplementary Information.

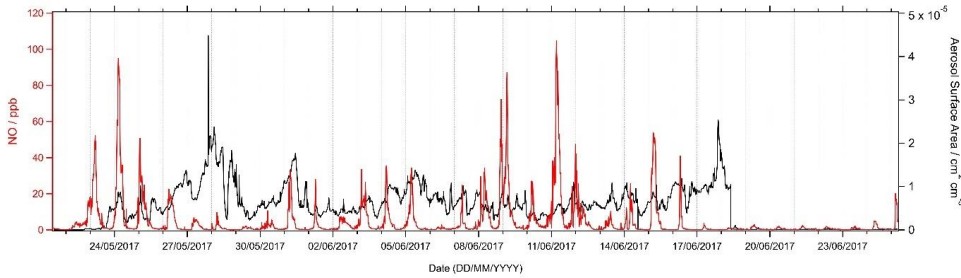

**Figure 3.** Time series of measured NO / ppb and $PM_{2.5}$ / $cm^2$ $cm^{-3}$ across entire summer AIRPRO campaign in Beijing.





## 3.2  Calculated $\gamma_{HO_2}$ for summer AIRPRO campaign

Measured values of [PM], copper (II) ion concentration and aerosol pH (used to calculate $H_{eff}$ in equation 1), and values of [ALWC] estimated using the ISORROPIA-II thermodynamic equilibrium model (Fountoukis and Nenes, 2007) were input into the parameterisation at a time resolution of 1 day. PM$_{2.5}$ mass concentration and Cu (II) ion concentration values were measured by extracting from filter samples offline with one filter sample taken every day. As such all measured values input into the parameterisation were averaged up to this time resolution. $R_d$ was calculated from the measured aerosol size distribution across the entire campaign. A value of 0.5 was chosen for the mass accommodation coefficient, $a_{HO_2}$, to reflect values previously measured for copper doped inorganic salts (Thornton and Abbatt, 2005; George et al., 2013; Taketani et al., 2008) and to allow better comparison with results from Song et al., 2020. For summer AIRPRO campaign, the soluble copper ion concentration was measured by extracting Cu (II) ions from filter samples and analysing the effluent using Inductively Coupled Plasma Mass Spectrometry (ICP-MS). As in Song et al., 2020, the total copper (II) mass concentration (ng m$^{-3}$ converted to g m$^{-3}$) was divided by the aerosol volume concentration (nm$^3$ cm$^{-3}$ converted to dm$^3$ m$^{-3}$) and the molar mass of copper (g mol$^{-1}$) to give the total copper molar concentration in the aerosol, [Cu$^{2+}$]$_{eff}$ (mol L$^{-1}$), which was then used in equation 1. The average values across summer AIRPRO campaign for parameters used in equation 1 are shown in Table 3.

| Parameter | Average value across campaign |
|---|---|
| **Temperature (K)** | 300 |
| **Relative humidity (%)** | 43 |
| **Aerosol pH** | 3 |
| **Count median radius (cm)** | $2.3\times10^{-6}$ |
| **ALWC (µg m$^{-3}$)**[a] | 14 |
| **[PM] (µg m$^{-3}$)** | 38.3 |
| **[Cu$^{2+}$]$_{eff}$ (mol L$^{-1}$)** | 0.0008 |
| **[Cu$^{2+}$]$_{eff}$ (ng m$^{-3}$)** | 4 |
| $a_{HO_2}$ | 0.5 (fixed) |

**Table 3.** Average values for summer AIRPRO campaign in Beijing, 2017 for parameters in equation 1. [a]This was a fixed minimum value of ALWC for the parameterisation to be used for this data set, fully explained in Section 3.4. Cu (II) ion concentration is given in both mol L$^{-1}$ and ng m$^{-3}$, due to mol L$^{-1}$ being used in equation 1 but ng m$^{-3}$ being the more atmospherically relevant unit.



For the Beijing summer AIRPRO campaign, an average value of $\gamma_{HO_2}$ = 0.07 ± 0.035 (1σ) was
calculated across the entire campaign, with values ranging from 0.002 to 0.15. The time series
for the calculated $\gamma_{HO_2}$, $R_d$ (cm), [PM] (µg m$^{-3}$), [ALWC] (µg m$^{-3}$) and [Cu$^{2+}$]$_{eff}$ (mol L$^{-1}$) is
shown in Figure 4.
As fully described in Song et al., 2020 supplementary information, the uncertainty in the
calculation of $\gamma_{HO_2}$ using equation 1 comes mainly from the uncertainty in [ALWC] (~10-20
%, calculated using ISORROPIA-II (Fountoukis and Nenes, 2007)), the uncertainty in the mass
accommodation coefficient (varying $a_{HO_2}$ within the parameterisation from 0.1 to 1, increased
the calculated $\gamma_{HO_2}$ from 0.042 to 0.077. However, by $a_{HO_2}$ = 0.5 this dependence has begun to
plateau with $\gamma_{HO_2}$ = 0.070 when $a_{HO_2}$ =0.5), and the uncertainty of the model calculations used
to formulate the parameterisation (~40 % as explained in Song et al., 2020). Uncertainties in
measured parameters i.e. temperature, [PM], [Cu$^{2+}$] and count median radius are due to
associated instrumental error which are assumed small in comparison.

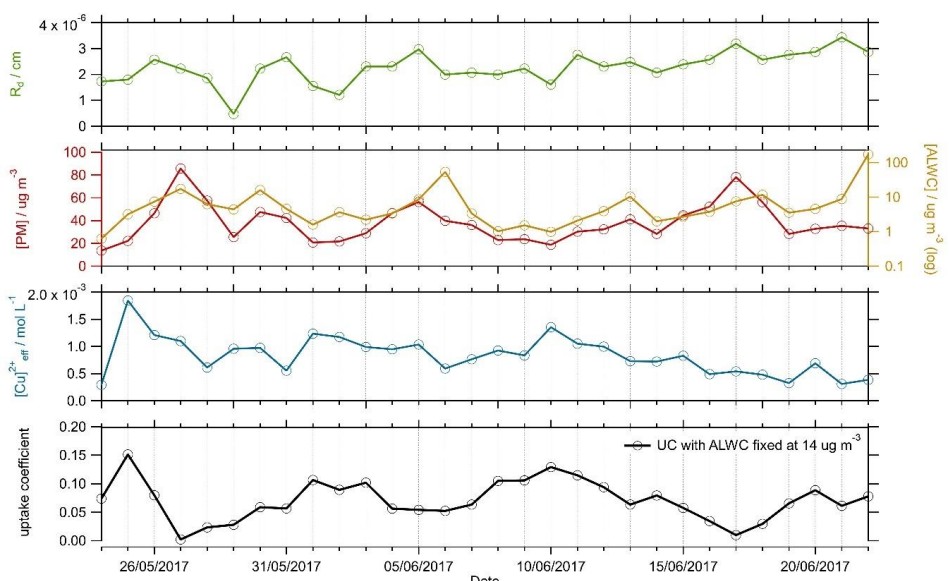

**Figure 4.** Time series of $R_d$ (cm, orange), [PM] (µg m$^{-3}$, red), [ALWC] (µg m$^{-3}$, yellow) and [Cu$^{2+}$]$_{eff}$ (mol L$^{-1}$, blue), parameters used in equation 1 to calculate $\gamma_{HO_2}$ (bottom panel). Each parameter has been averaged up to a time resolution of 1 day to match the lowest resolution measurement. The calculated $\gamma_{HO_2}$ is shown in the bottom panel, for a fixed [ALWC] = 14 ug m$^{-3}$ (solid black line).

To examine the effect within the Song parameterisation of [PM] and [ALWC] on $\gamma_{HO_2}$ as a
function of copper molarity, the uptake coefficient was calculated by varying the [Cu$^{2+}$]$_{eff}$





concentration within the parameterisation with alternatively fixed values of [PM] or [ALWC].
For a given value of $[Cu^{2+}]_{eff}$, at fixed [ALWC] an increase in [PM] causes a decrease in the
curvature of $\gamma_{HO_2}$ vs $[Cu^{2+}]_{eff}$, whereas at a fixed [PM], an increase in [ALWC] leads to an
increase in $\gamma_{HO_2}$ for a given $[Cu^{2+}]_{eff}$. As shown in Figure 5, [AWLC] and [PM] have the
greatest effect on $\gamma_{HO_2}$ between $[Cu^{2+}]_{eff}= 10^{-5}$-$10^{-1}$M before the curve levels off towards the
mass accommodation coefficient of 0.5, as input into the model. For context within the Beijing
campaign, the curve of $\gamma_{HO_2}$ vs $[Cu^{2+}]_{eff}$ was plotted in Figure 5 using the average values for
the AIRPRO summer campaign fixed at [ALWC] = 14 ug m$^{-3}$ and [PM] = 38.3 ug m$^{-3}$. For the
average AIRPRO summer campaign values, an increase $[Cu^{2+}]_{eff}$ has the most effect on $\gamma_{HO_2}$
between $[Cu^{2+}]_{eff}$ ~$10^{-3}$ -$10^{-1}$ M, with the average $[Cu^{2+}]_{eff}$ for the campaign being $8 \times 10^{-4}$ M
(values ranged from $3 \times 10^{-4}$ to $2 \times 10^{-3}$ M across campaign).







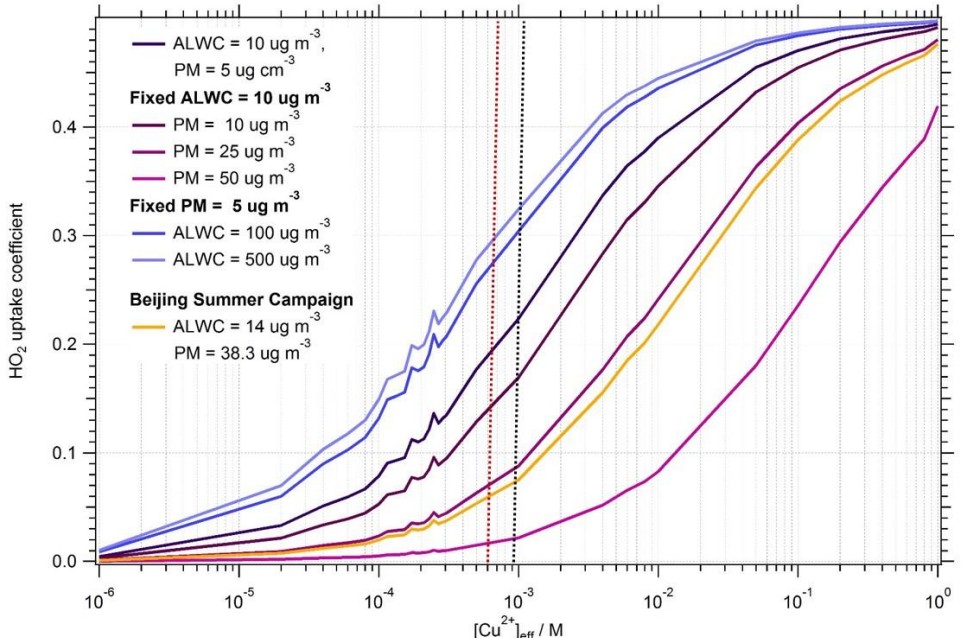

**Figure 5.** Dependence of uptake coefficient, $\gamma_{HO_2}$ on aerosol copper concentration, $[Cu^{2+}]_{eff}$ (M), showing the effect of varying [PM] with fixed [ALWC] and vice versa. Pink to purple lines show the effect on uptake coefficient of varying [PM] from 5-50 µg m⁻³ with a fixed [ALWC] of 10 g cm⁻³. Blue to dark blue lines show the effect on $\gamma_{HO_2}$ of varying [ALWC] from 10-500 ug m⁻³ (much higher than typically seen atmospherically) with a fixed [PM] of 5 µg m⁻³. The yellow line shows the effect on the $\gamma_{HO_2}$ of varying $[Cu^{2+}]_{eff}$, with [ALWC] and [PM] taken as the averages from the Beijing campaign, i.e. [ALWC] = 14 ug m⁻³ and [PM] = 38.8 ug m⁻³. Black dashed line indicates the average $[Cu^{2+}]_{eff}$ for Beijing summer campaign. Red dashed line indicates the average $[Cu^{2+}]_{eff}$ for the Wangdu campaign. Note that the [PM] and [ALWC] are both higher for Wangdu campaign compared to the Beijing campaign.

## 3.3 Box modelling results

### 3.3.1 Effect of calculated $\gamma_{HO_2}$ on modelled AIRPRO Summer radical concentrations

As reported in Whalley et al., 2021, radical concentrations were high during the AIRPRO summer campaign with maximum measured concentrations of OH, HO$_2$ and RO$_2$ of $2.8 \times 10^7$ molecule cm⁻³, $1 \times 10^9$ molecule cm⁻³ and $5.5 \times 10^9$ molecule cm⁻³ on the afternoons of the 30[th] May, 9[th] June and 15[th] June respectively. The time series of measured OH, HO$_2$ and RO$_2$ for the entire summer campaign as measured by the Leeds FAGE instrument with MCM_base model outputs for OH, HO$_2$ and RO$_2$ can be found in Whalley et al., 2021. Using the MCM and the $\gamma_{HO_2}$ calculated for the Summer Beijing campaign with the Song parameterisation, the





effect of HO$_2$ uptake on the concentration of OH, HO$_2$ and RO$_2$ radicals was investigated and
compared to the base model.
The MCM_base model predicted radical concentrations are shown as average diurnal profiles
compared to both the measured diurnals and the MCM_gamma model in Figure 6. A detailed
description of the diurnal variation in measured and modelled OH, HO$_2$ and RO$_2$ radicals for
the summer Beijing campaign is given in Whalley et al., 2021, so only a brief summary will be
given here.
The average diurnal profiles show that the MCM_base model can re-produce the measured OH
concentrations relatively well, however the modelled peak in OH is shifted to the afternoon
with a peak at ~14:00 compared to the midday peak in the observations. In comparison, HO$_2$
is over-predicted, particularly during the day with the exception being when NO was high from
9-12$^{th}$ June. Day-time HO$_2$ is over-predicted on average by MCM_base by up to a factor of
~2.9 with a peak in the diurnal at ~ 14:30. In-comparison, daytime RO$_2$ concentration is under-
predicted on average by MCM_base by up to a factor of ~7.5, with a larger under-prediction
in the morning between ~6:30-10:30 when NO levels were highest. At the peak of the RO$_2$
diurnal, on average the concentration was under-predicted by MCM_base by a factor of ~2.7.
While the MCM_base model is able to reproduce measured OH concentrations reasonably
well, the inability of this model to reproduce HO$_2$ and RO$_2$ suggests missing key reactions. In
Whalley et al., 2021, budget analysis highlighted a missing source of OH, in addition to a
missing RO$_2$ production reaction which could partially explain the under-prediction of RO$_2$ by
the MCM_base model. It was also suggested that the over-prediction of HO$_2$ could be due, in
part, to an under-prediction in the rate of reaction of RO$_2$ with NO to produce a different RO$_2$
species, i.e. RO$_2$+NO→RO$_2$', which would lead to propagation of RO$_2$ to different, more
oxidised RO$_2$ species, competing with the recycling of RO$_2$ via RO to give HO$_2$, or due to lack
of RO$_2$ autoxidation pathways within the model which could lead to the formation of highly
oxygenated molecules as opposed to HO$_2$. The higher measured RO$_2$ concentrations could,
therefore, suggest that the lifetime of total RO$_2$ is longer than currently considered in the model.
As stated in Section 3.3, for the Beijing summer AIRPRO campaign, values of calculated $\gamma_{HO_2}$
varied ranging from 0.002 to 0.15, giving an average value of $\gamma_{HO_2} = 0.07 \pm 0.035$ (1σ) across
the campaign. These $\gamma_{HO_2}$ values calculated on a daily time resolution, were added into the
MCM_base model to give the MCM_gamma model. The average median diurnals of modelled




OH, $HO_2$ and $RO_2$ (molecule cm$^{-3}$) for MCM_base, MCM_gamma (with $\gamma_{HO_2}$ ranging from
0.002-0.15) and MCM_SA (with $\gamma_{HO_2}$ fixed at 0.2) are shown in Figure 6.

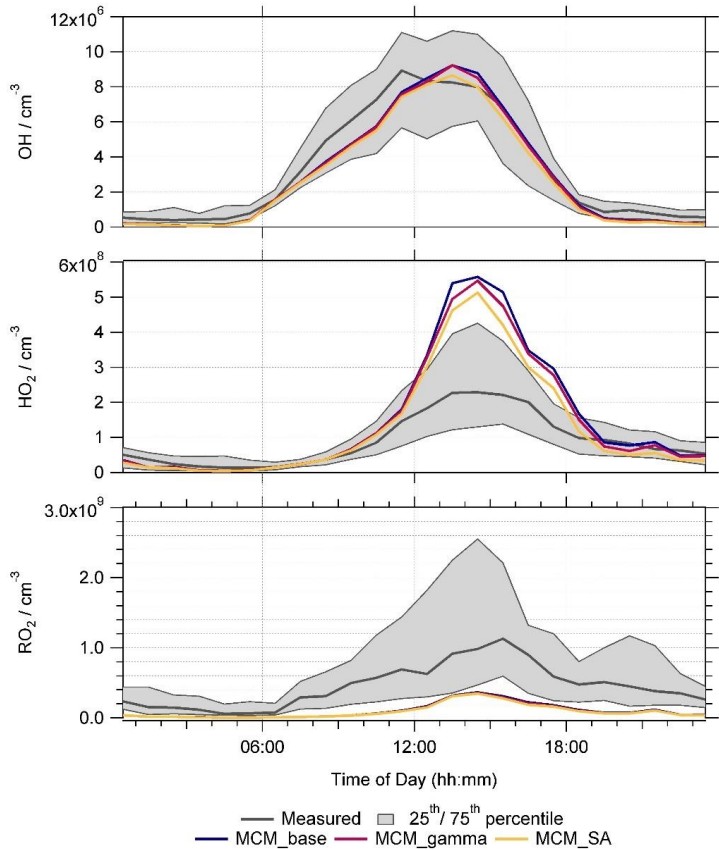

**Figure 6.** Average median diurnals for measured radical concentrations (grey) and modelled OH, $HO_2$ and total $RO_2$ radical concentrations in molecule cm$^{-3}$ for MCM_base (blue), MCM_gamma (dark pink) and MCM_SA (yellow) model runs. All diurnal's are 60 minute averages, taken over the entire measurement period. Shaded grey regions represent the 25$^{th}$/75$^{th}$ percentiles of measured radical data.

Due to a combination of the calculated uptake coefficient being smaller, on average, than
usually used within models (i.e < 0.2), and the high $NO_x$ levels, little effect on average radical
diurnals was seen by adding in $HO_2$ aerosol uptake into the model. Figure 6 shows that the OH
and $RO_2$ radical concentrations were not significantly affected on average across the campaign
by the addition of aerosol uptake. The average median diurnal of $HO_2$ can be seen as slightly
decreased, i.e. the over-prediction of $HO_2$ is slightly less for MCM_gamma compared to
MCM_base, with the over-prediction decreasing from a factor of ~2.9 to ~2.4 at the 14:30 peak
in the diurnal.





Due to the recycling of $RO_2$ to $HO_2$ and then back to OH by NO, it is important to consider the
dependency of radicals on NO and whether the addition of the $HO_2$ uptake coefficient has an
effect on the model's ability to predict the dependency of radical concentrations on NO. The
dependency of measured/modelled OH, $HO_2$ and $RO_2$ on NO mixing ratio is discussed fully
for the MCM_base model in Whalley et al., 2021. Figure 7 shows the ratio of measured to
modelled OH, $HO_2$ and $RO_2$ radical concentrations binned against NO mixing ratio (ppb) for
MCM_gamma, compared to MCM_base.

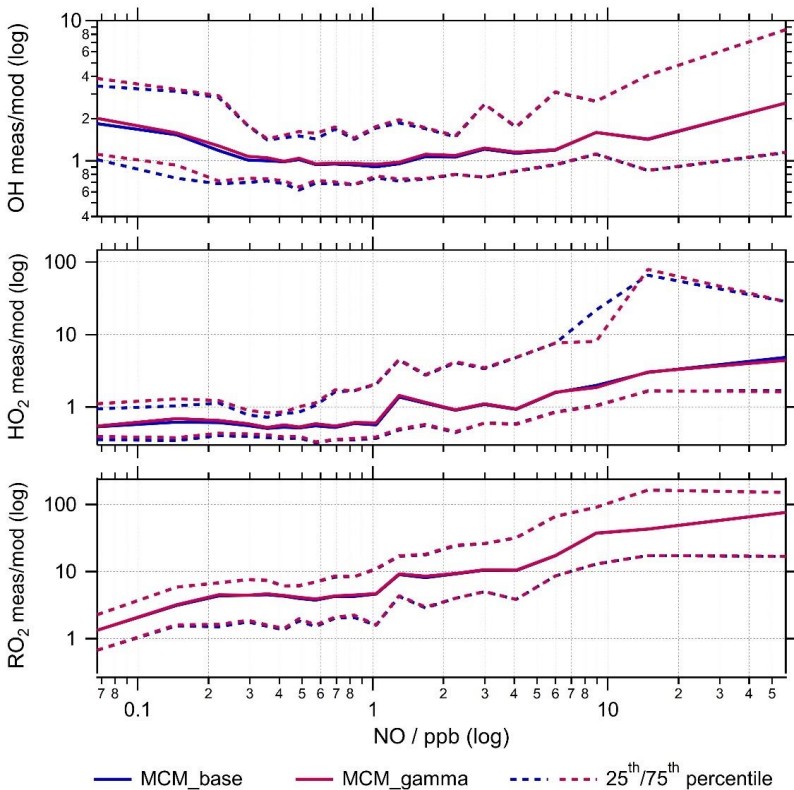

**Figure 7.** Ratio of measured to modelled OH, $HO_2$ and $RO_2$ radical concentrations using the MCM_base (blue) and MCM_gamma (dark pink) model binned over the range of NO mixing ratios (ppb) for the summer AIRPRO campaign. Solid lines show the median average measured to modelled radical concentration. Dashed lines show the 25th/75th percentiles.

For the range of NO mixing ratios observed across the summer AIRPRO campaign, the OH
measured to modelled ratio is close to 1 between ~0.3 and 2 ppb NO with the MCM_base
model beginning to under-predict OH slightly both below 0.3 ppb NO and above 2 ppb NO.
Both $HO_2$ and $RO_2$ radical concentrations were strongly dependent on NO mixing ratio, with





the model over-predicting $HO_2$ below ~ 1 ppb NO. For the entire campaign the average NO
was 4.7 ppb with 45% of NO measurements taken across campaign being less than or equal to
1 ppb. Across all NO mixing ratios the measured to modelled ratio for $RO_2$ shows a large
under-prediction, with the largest under-prediction at the highest NO mixing ratios. This is
likely contributing to the underprediction of $HO_2$ at higher NO mixing ratios. From Figure 7 it
can be seen that the addition of the calculated $HO_2$ uptake coefficient has had little effect across
the range of NO mixing ratios measured during the summer AIRPRO campaign.
To showcase any effect adding $HO_2$ aerosol uptake would have on $HO_2$ loss pathways as a
whole, and thereby make a judgement on the effect of decreased $PM_{2.5}$ and hence $HO_2$ loss via
aerosol surfaces on the $O_3$ production within Beijing, a rate of destruction analysis (RODA)
was done for MCM_gamma. The loss pathways of $HO_2$ within MCM_gamma are shown in
Figure 8 as an average median diurnal and as a function of NO mixing ratio (ppb), in addition
to the percentage contribution of $HO_2$ uptake to the overall loss of $HO_2$ within the model.

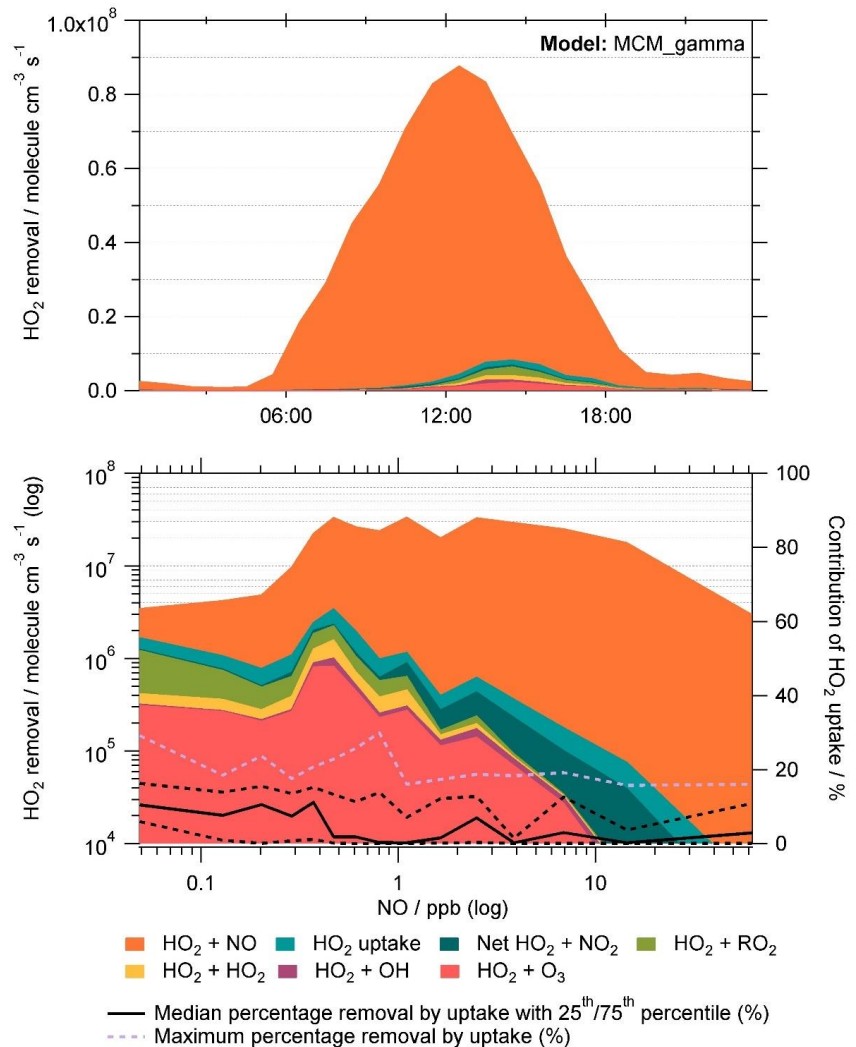

**Figure 8.** Rate of destruction analysis (RODA) showing the dominant loss pathways of $HO_2$ within MCM_gamma shown as (a) a diurnal variation and (b) as a function of NO mixing ratio (ppb). Median removal of $HO_2$ by uptake (%) as a function of NO (ppb) is shown as solid black line in (b), with 25th/75th percentile shown as the black dashed lines. Maximum percentage removal by uptake for a given NO mixing ratio is shown as a lilac dashed line.

As shown in the RODA, the dominant loss pathway of $HO_2$ is $HO_2$ + NO across the entire
campaign (90 ± 14 % of total loss), followed by $HO_2$ + $RO_2$ (3.5 ± 8.1 % of total loss). This is
expected due to high levels of $NO_x$ in Beijing, especially during the day. As seen in the RODA
diurnal, the $HO_2$ + NO loss pathway peaks at midday following the morning peak in NO mixing
ratio due to rush hour traffic. As NO mixing ratio decreases, the relative importance of other





loss pathways of $HO_2$ increases. At the lowest NO mixing ratio, i.e. < 0.1 ppb NO, the loss
pathways of $HO_2$ within MCM_gamma with the largest contribution to total loss were $HO_2$ +
NO (55 ± 19 %), $HO_2$ + $RO_2$ (23 ± 17 %) and $HO_2$ + $O_3$ (9.3 ± 4.1 %). It is worth noting that
as the NO mixing ratio decreases the relative importance of $HO_2$ removal by $O_3$ increases
presumably due to the titration reaction of $O_3$ with NO decreasing (and hence higher observed
$[O_3]$). This could be important when considering policy changes with $NO_x$ pollution in China
decreasing in recent years. The contribution of the various loss pathways of $HO_2$ to total $HO_2$
loss within MCM_gamma under low (< 0.1 ppb) and high (>0.1 ppb) NO are compared in
Table 4.

|  | $HO_2+O_3$ | $HO_2+OH$ | $HO_2+HO_2$ | $HO_2+RO_2$ | Net $HO_2+NO_2$ | $HO_2+NO$ | uptake |
|---|---|---|---|---|---|---|---|
| **Low NO (< 0.1 ppb)** | 9.3 ± 4.1 | 0.1 ± 0.1 | 3.0 ± 1.8 | 23 ± 17 | 2.4 ± 3.0 | 55 ± 19 | 7.3 ± 7.3 |
| **High NO (> 0.1 ppb)** | 1.8 ± 2.3 | 0.2 ± 0.3 | 0.8 ± 1.3 | 2.0 ± 4.4 | 0.4 ± 1.2 | 93 ± 9.0 | 1.9 ± <0.01 |

**Table 4.** Average relative percentage contribution of individual $HO_2$ loss pathways to the total loss of $HO_2$ within
MCM_gamma, averaged for days when NO was low, (< 0.1 ppb) and high (> 0.1 ppb). Net $HO_2+NO_2$ refers to
$HO_2+NO_2 \rightarrow HO_2NO_2$ minus $HO_2NO_2 \rightarrow HO_2+NO_2$.
Though there is not a strong dependence of $HO_2$ aerosol uptake loss pathway on NO mixing
ratio for the calculated $\gamma_{HO_2}$ (av. 0.07 ± 0.035) within MCM_gamma, it can be seen that at the
lowest NO mixing ratios an average of ~7 % of total $HO_2$ loss is due to uptake, with a maximum
at the lowest NO of ~29% (shown as lilac dashed line in Figure 8). This is a significant loss of
$HO_2$, especially on days where the NO mixing ratio is low and the aerosol surface area is high,
highlighting that the uptake of $HO_2$ onto aerosols could be important, and will be increasingly
so at lower NO.
**3.3.2   Comparison to $\gamma_{HO_2}$ fixed at 0.2**
While the maximum $\gamma_{HO_2}$ calculated using the Song parameterisation for the summer AIRPRO
campaign was 0.15, to provide context with previous modelling studies, the commonly used
fixed value of $\gamma_{HO_2}= 0.2$ was added into the MCM_base model to give the MCM_SA model.
The average median diurnals of modelled OH, $HO_2$ and $RO_2$ (molecule cm$^{-3}$) for MCM_base,
MCM_gamma and MCM_SA are shown in Figure 6.
In comparison to calculated $\gamma_{HO_2}$ in MCM_gamma, a fixed $\gamma_{HO_2}= 0.2$ had a more significant
effect on radical concentrations. While the median diurnal shows that the $RO_2$ concentration
was not significantly affected by the addition of $HO_2$ uptake, the over-prediction seen in the
average median $HO_2$ concentration compared to the measurements at the 14:30 peak decreased





from a factor of ~2.9 in MCM_base to ~2.3. OH radical concentrations were still relatively
well reproduced with early afternoon OH concentrations predicted better though this is due to
a shift in the modelled peak compared to the measured concentration peaking at midday.
As seen in Figure 9, the addition of $\gamma_{HO_2} = 0.2$ affected the ability of the model to reproduce
the NO dependence of radical concentrations. While MCM_base over-predicts $HO_2$ below ~ 1
ppb NO, the over-prediction of $HO_2$ decreases below 1 ppb NO for MCM_SA with $HO_2$ being
well reproduced at the lowest NO mixing ratios (i.e. < 0.1 ppb) due to the relative increase in
the importance of $HO_2$ uptake as a sink of $HO_2$. Modelled $RO_2$ is not significantly affected by
the addition of $HO_2$ uptake at any NO mixing ratio. The modelled concentration of OH is under-
predicted for the entire range of NO mixing ratios compared to measured values, though only
slightly between ~ 1 and 6 ppb NO. Below ~ 4 ppb NO, the underprediction of OH by
MCM_SA increases compared to MCM_base due most likely to loss of $HO_2$ onto aerosols
competing with loss via NO to give OH. Budget analysis done by Whalley et al., 2021,
showcases that with a reduction in over-prediction of modelled $HO_2$, OH is under-predicted
revealing a missing OH source.



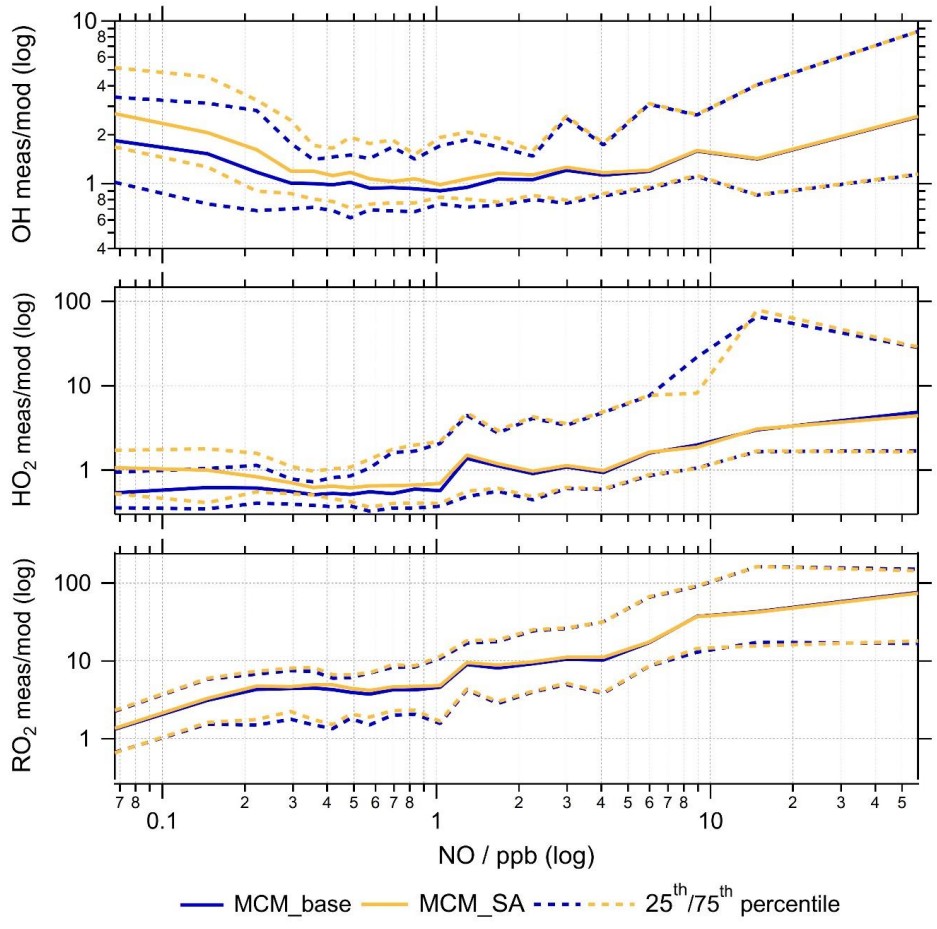

**Figure 9.** Ratio of measured to modelled OH, HO$_2$ and RO$_2$ radical concentrations using the MCM_base (blue) and MCM_SA (yellow) model binned over the range of NO mixing ratios (ppb) for the summer AIRPRO campaign. Solid lines show the median average measured to modelled radical concentration. Dashed lines show the 25th/75th percentile.

557 Analysis of the RODA for MCM_SA shows that with $\gamma_{HO_2} = 0.2$ HO$_2$ aerosol uptake is a

558 significant contributor to total loss of HO$_2$ (8.1 ± 13 %, averaged for all NO mixing ratios).

559 However, for all NO mixing ratios HO$_2$ + NO is still the dominant loss pathway (86 ± 18 %),

560 as expected. At the lowest NO mixing ratios (i.e. < 0.1 ppb) an average of ~29 % of total HO$_2$

561 loss is due to uptake, with a maximum at the lowest NO of ~78%, shown in Figure 10. The

562 contribution of the various loss pathways of HO$_2$ to total HO$_2$ loss within MCM_gamma under

563 low (< 0.1 ppb) and high (>0.1 ppb) NO are compared in Table 5. The comparison of





percentage contribution of HO$_2$ uptake to total HO$_2$ removal binned against NO mixing ratio
(ppb) for MCM_gamma and MCM_SA RODA is shown in Figure 10.

|  | HO$_2$+O$_3$ | HO$_2$+OH | HO$_2$+HO$_2$ | HO$_2$+RO$_2$ | Net HO$_2$+NO$_2$ | HO$_2$+NO | uptake |
|---|---|---|---|---|---|---|---|
| **Low NO (< 0.1 ppb)** | $6.9 \pm 3.5$ | $0.1 \pm 0.1$ | $1.7 \pm 1.4$ | $17 \pm 14$ | $1.6 \pm 2.2$ | $44 \pm 24$ | $29 \pm 24$ |
| **High NO (> 0.1 ppb)** | $1.8 \pm 2.1$ | $0.2 \pm 0.2$ | $0.6 \pm 1.0$ | $1.7 \pm 3.8$ | $0.4 \pm 1.0$ | $89 \pm 13$ | $6.5 \pm 9.7$ |

**Table 5.** Average relative percentage contribution of individual HO$_2$ loss pathways to the total loss of HO$_2$ within
MCM_SA (fixed $\gamma_{HO_2} = 0.2$), averaged for days when NO was low, (< 0.1 ppb) and high (> 0.1 ppb). Net
HO$_2$+NO$_2$ refers to HO$_2$+NO$_2$→HO$_2$NO$_2$ minus HO$_2$NO$_2$→HO$_2$+NO$_2$.

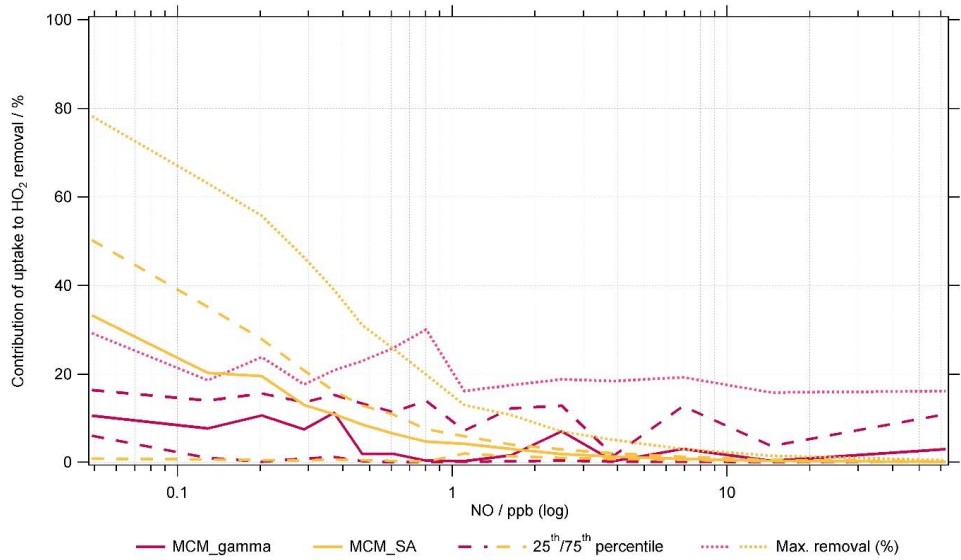

**Figure 10.** Average percentage contribution of HO$_2$ uptake to total HO$_2$ removal within MCM_gamma (pink
line, $\gamma_{HO_2} = 0.070 \pm 0.035$) and MCM_SA model (yellow line, $\gamma_{HO_2} = 0.2$) for Summer AIRPRO campaign
plotted as a function of NO mixing ratio (ppb). Dashed lines represent the 25$^{th}$/75$^{th}$ percentiles. Dotted lines
represent maximum removal.

### 3.3.3 Effect of $\gamma_{HO_2}$ on the O$_3$ regime

### 3.3.3.1 Calculation of L$_N$/Q and absolute O$_3$ sensitivity

First introduced by Kleinman et al., 1997, L$_N$/Q is the ratio of radical loss via NO$_x$ to total
primary radical production and is used as a means of determining O$_3$ production sensitivity to
VOCs and NO$_x$ (Kleinman, 2000; Kleinman et al., 1997; Kleinman et al., 2001). This method
was then built on by Sakamoto et al., 2019 who included loss of peroxy radicals
(XO$_2$=HO$_2$+RO$_2$) onto aerosol surfaces within the calculation of O$_3$ sensitivity.





The only source of tropospheric $O_3$ is by the reaction of peroxy radicals with NO, while the
main source of $XO_2$ species is via the reaction of OH with VOCs.

$$XO_2 + NO \rightarrow XO + NO_2 \tag{R 1}$$

$$OH + VOC + O_2 \rightarrow XO_2 + products \tag{R 2}$$

The $O_3$ production rate in the troposphere is therefore:

$$P(O_3) = k_{HO_2+NO}[HO_2][NO] + k_{RO_2+NO}[RO_2][NO] \tag{2}$$

where $k_{HO_2+NO}$ and $k_{RO_2+NO}$ are the bimolecular rate constants for the reaction of $HO_2$ and
$RO_2$ with NO.
The production rate of OH, $HO_2$ and $RO_2$ radicals, Q, must equal the loss rate:

$$Q = L_P + L_N + L_R \tag{3}$$

where $L_P$ is the loss rate of radicals onto aerosol particles, $L_N$ is the loss rate of radicals via
reaction with $NO_x$ species and $L_R$ is the loss rate of radicals via radical-radical reactions to give
peroxides.

$$L_P = k_{HO_2\ uptake}[HO_2] + k_{RO_2\ uptake}[RO_2] = k_P[XO_2] \tag{4}$$

$$L_N \approx k_{NO_2+OH}[NO_2][OH] \tag{5}$$

$$L_R = 2(k_{HO_2+HO_2}[HO_2]^2 + k_{RO_2+HO_2}[HO_2][RO_2]) \tag{6}$$

where $k_{HO_2\ uptake}$ is the rate constant for the loss of $HO_2$ onto aerosol surfaces, $k_{RO_2\ uptake}$ is
the rate constant for the loss of $RO_2$ onto aerosol surfaces, $k_{NO_2+OH}$ is the bimolecular rate
constant for the reaction of $NO_2$ with OH, $k_{HO_2+HO_2}$ is the bimolecular rate constant for the
self-reaction of $HO_2$ and $k_{RO_2+HO_2}$ is the bimolecular rate constant for the reaction of $RO_2$ with
$HO_2$.
For radical loss onto aerosol surfaces, the rate constant is given as a function of the reactive
uptake coefficient, $\gamma_{XO_2}$, aerosol particle surface area ($cm^2\ cm^{-3}$) and mean thermal velocity
($cm\ s^{-1}$), given by $v = \sqrt{8RT/\pi M}$ with R, T and M as the gas constant, the absolute temperature
and the molar mass of species respectively.

$$k_{radical\ uptake} = \frac{\gamma_{XO_2} \times SA \times v}{4} \tag{7}$$

According to the method described in Sakamoto et al., 2019, the ratio of radical loss to $NO_x$ to
primary $O_3$ production including radical loss via aerosol uptake, $\frac{L_N}{Q}$ is defined as follows:





$$\frac{L_N}{Q} = \frac{1}{1 + \left( \dfrac{(2k_R[XO_2] + k_P)k_{OH+VOC}[VOC]}{(1-\alpha)k_{HO_2+NO}[NO]k_{NO_2+OH}[NO_2]} \right)} \tag{8}$$

where $k_{OH+VOC}$ is the bimolecular rate constant for the loss of OH via reaction with VOCs and
$(1 - \alpha)$ is the fraction of $XO_2$ that is $HO_2$.
The relative sensitivity of $O_3$ production to $NO_x$ and VOCs is described by:

$$\frac{\delta lnP(O_3)}{\delta ln\,[NO_x]} = (1-\chi)\left( \frac{1 - \dfrac{3}{2}\dfrac{L_N}{Q}}{1 - \dfrac{1}{2}\dfrac{L_N}{Q}} \right) + \chi\left(1 - 2\frac{L_N}{Q}\right) \tag{9}$$

$$\frac{\delta lnP(O_3)}{\delta ln\,[VOC]} = (1-\chi)\left( \frac{\dfrac{1}{2}\dfrac{L_N}{Q}}{1 - \dfrac{1}{2}\dfrac{L_N}{Q}} \right) + \chi\frac{L_N}{Q} \tag{10}$$

where $\chi = \frac{L_P}{L_P + L_R}$. The $O_3$ regime transition point, where $\frac{\delta lnP(O_3)}{\delta ln\,[NO_x]} = \frac{\delta lnP(O_3)}{\delta ln\,[VOC]}$, is given by $\frac{L_N}{Q_{trans}}$.

$$\frac{L_N}{Q_{trans}} = \frac{1}{2}(1 - \chi) + \frac{1}{3}\chi \tag{11}$$

Absolute $O_3$ sensitivity was introduced by Sakamoto et al., 2019, and allows for the assessment
of how reduction in $O_3$ precursors could contribute to reduction in $P(O_3)$ by integrating over
time and area. The absolute sensitivity of $O_3$ production to VOC and $NO_x$ is then described by:

$$Absolute\ P(O_3) = \frac{\delta P(O_3)}{\delta ln\,[X]} = P(O_3)\frac{\delta P(O_3)}{\delta ln\,[X]} \tag{12}$$

where is *[X]* is $NO_x$ or VOC.
$\frac{L_N}{Q}$ was calculated for all model runs, MCM_base, MCM_gamma and MCM_SA using
modelled [$HO_2$] and [$RO_2$] concentrations but measured values of [NO] and [$NO_2$], to
investigate the effect on the $O_3$ regime of adding $HO_2$ aerosol uptake into the model. The time
series of calculated $\frac{L_N}{Q}$ for all models, in addition to the regime transition point, $\frac{L_N}{Q_{trans}}$ for the
entire campaign is shown in Figure 11.
When $\frac{L_N}{Q} < \frac{L_N}{Q_{trans}}$, this is defined as a $NO_x$-sensitive regime, meaning that small changes in $NO_x$
will affect the rate of in situ $O_3$ production. This can be seen on a few days across the campaign,
specifically in the afternoon, due to $NO_x$ peaking in the morning due to traffic emissions before
rapidly decreasing in the afternoon which pushes the $O_3$ regime on certain days from VOC-





limited to NO$_x$-limited. However, for the majority of the campaign, the O$_3$ production regime
is VOC-limited, for all models, meaning that O$_3$ production rates will not be significantly
affected by small changes in NO$_x$.
Binning $\frac{L_N}{Q}$ against NO mixing ratio (ppb), in Figure 12, shows the change from VOC to NO$_x$-
limited regime at very low NO mixing ratios for MCM_base, MCM_gamma and MCM_SA.
As aerosol uptake is added the transition to NO$_x$-limited regime occurs at higher NO, with
average median transition point occurring at ~ 0.2 ppb NO for MCM_gamma (average $\gamma_{HO_2}$=
0.070 ± 0.035) and at ~0.5 ppb NO for MCM_SA (fixed $\gamma_{HO_2}$= 0.2). This suggests that a
reduction in PM (and therefore uptake of HO$_2$ onto aerosols) would delay the transition to a
NO$_x$-sensitive regime until lower NO$_x$ levels are reached. Therefore, any emissions policy
aimed at reduced NO$_x$ to decrease O$_3$ levels would not be as effective if PM is decreasing at
the same time.

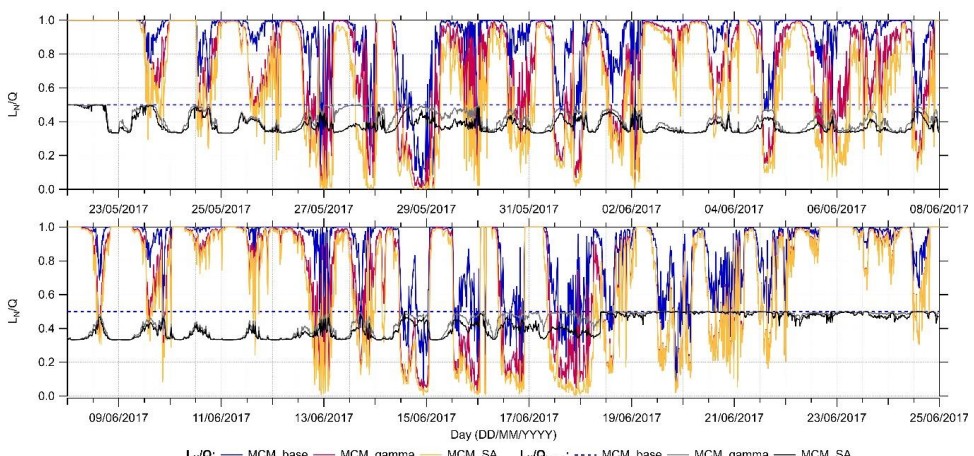

**Figure 11.** Time series of calculated $\frac{L_N}{Q}$ and $\frac{L_N}{Q_{trans}}$ values for MCM_base (blue), MCM_gamma (pink) and

MCM_SA (yellow) models across the entire summer AIRPRO campaign. $\frac{L_N}{Q_{trans}}$ for MCM_gamma is shown as

grey line, while $\frac{L_N}{Q_{trans}}$ for MCM_SA is the black line.




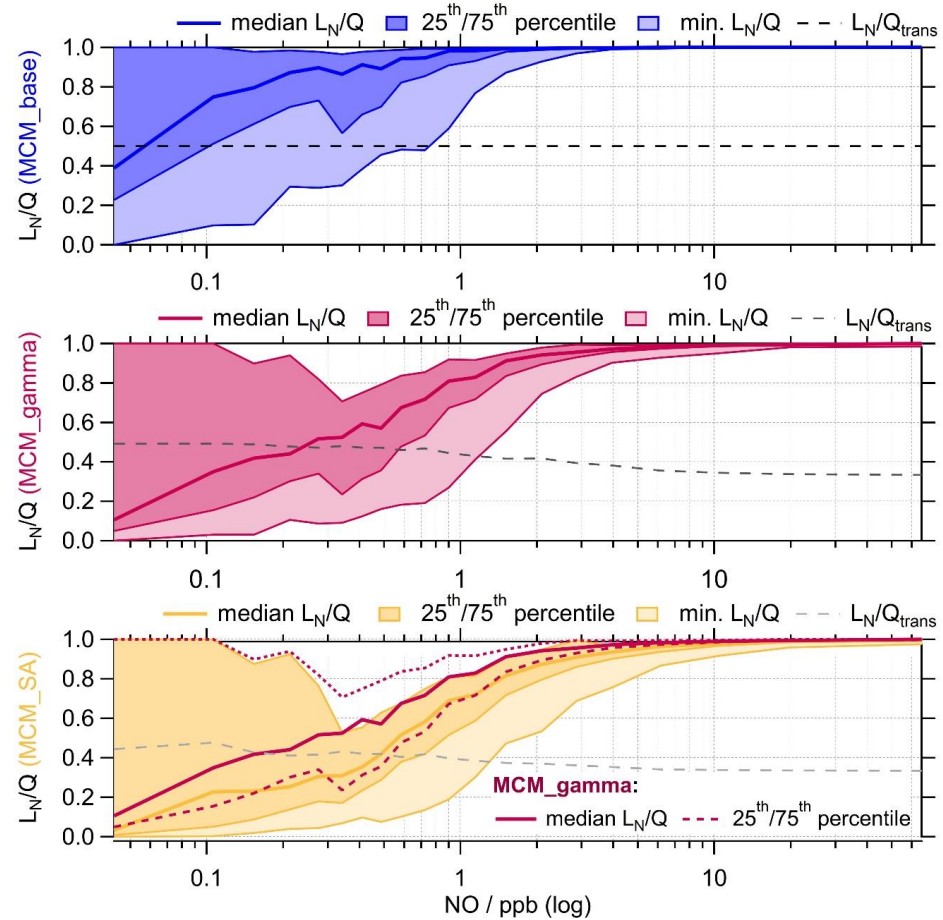

**Figure 12.** $\frac{L_N}{Q}$ for MCM_base (blue, top panel), MCM_gamma (pink, middle panel) and MCM_SA (yellow, bottom panel) binned against the log of measured NO mixing ratio for the entire summer AIRPRO campaign. $\frac{L_N}{Q_{trans}}$ for MCM_base (black dashed line) taken as 0.5 for entire range of NO mixing ratios. $\frac{L_N}{Q_{trans}}$ for MCM_gamma (dark grey dashed line) and MCM_SA (light grey dashed line) calculated using equation 11. 25th/75th percentiles and minimum $\frac{L_N}{Q}$ are plotted to show full spread of data for each model scenario.

The average median diurnal of absolute P(O$_3$), $\frac{\delta P(O_3)}{\delta \ln [X]}$ , for the MCM_gamma and MCM_SA
over the entire campaign is shown in Figure 13. The time series of absolute P(O$_3$), averaged up
to a daily time resolution, across the entire measurement period can be found in Supplementary
Information as SI Figure 2.

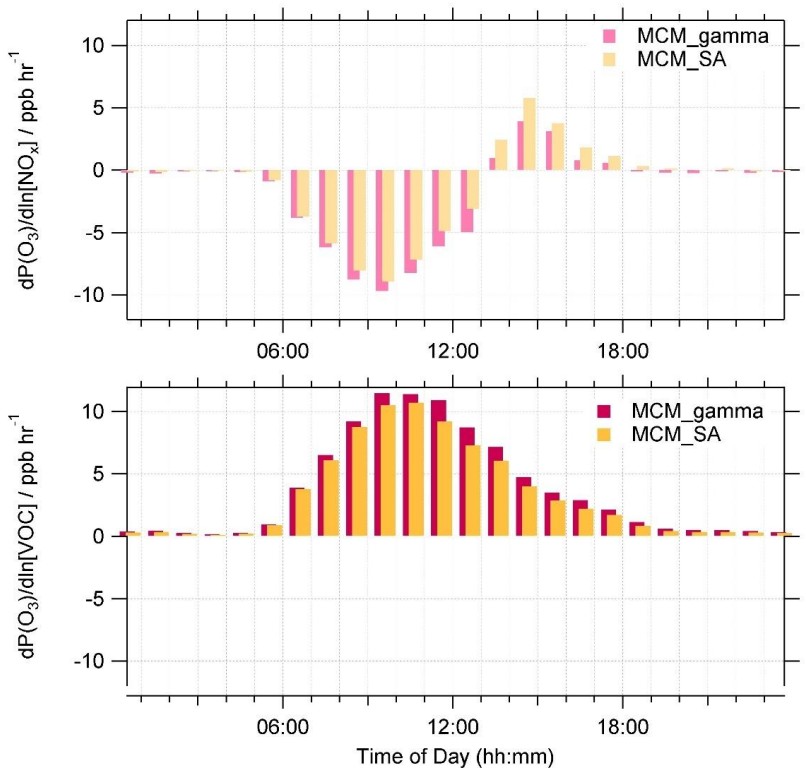

**Figure 13.** Average median diurnal of absolute $O_3$ sensitivity to $NO_x$ (top panel) and VOC (bottom panel) in ppbV h$^{-1}$ for MCM_gamma (pink) and MCM_SA (yellow) across the entire summer AIRPRO campaign. MCM_gamma includes $\gamma_{HO_2}$ calculated using the Song parameterisation (av. $0.070 \pm 0.035$) while MCM_SA includes $\gamma_{HO_2}$ at a fixed value of 0.2. All diurnals are 60 minute averages.

As expected from $\frac{L_N}{Q}$ calculations, calculations of absolute $O_3$ production sensitivity showcase
that for both MCM_gamma and MCM_SA, on average, the $O_3$ regime was VOC sensitive
throughout the day with $NO_x$ sensitivity increasing in the afternoons. On a few days, when low
NO mixing ratio coincided with high SA, the $O_3$ regime can be seen shifting from VOC to $NO_x$
limited. An example of this can be found in SI Figure 3, for the 17/06/2017 and 18/06/2017
when the average NO mixing ratio was $0.41 \pm 0.50$ ppb and the average SA was $(8.4 \pm 6.2) \times$
$10^{-6}$ cm$^2$ cm$^{-3}$. With an increase in $\gamma_{HO_2}$ between MCM_gamma and MCM_SA, the sensitivity
of $O_3$ regime to VOC decreased but sensitivity to $NO_x$ increased. This effect could be important
for areas where $O_3$ production regime is $NO_x$ sensitive or less strongly VOC sensitive. With
$NO_x$ levels reportedly decreasing across China in recent years (Krotkov et al., 2016; Liu et al.,
2016; Miyazaki et al., 2017; Van Der A et al., 2017), $O_3$ production regimes would be expected
to move more towards $NO_x$-sensitive regimes in urban China. However, with concomitant





reduction in PM (Ma et al., 2016b; Lin et al., 2018), this transition to a $NO_x$-sensitive regime
may be delayed until lower $NO_x$ levels are reached.
Our result for the Beijing campaign are consistent with the results of Song et al., 2022 which
concluded that for the conditions of the Wangdu campaign the addition of $HO_2$ uptake does not
change the overall $O_3$ sensitivity regime throughout the campaign. However, the shift in $O_3$
sensitivity regime from VOC-limited to $NO_x$-limited from the consideration of $HO_2$ uptake
could be important for areas with lower $NO_x$ and high aerosol particle loading.

## 4    Conclusions

Using the Song parameterisation, the heterogeneous uptake coefficient of $HO_2$, $\gamma_{HO_2}$, was
calculated for the summer AIRPRO campaign in Beijing, 2017 as a function of measured
$[Cu^{2+}]_{eff}$, [ALWC] and [PM]. The calculated average $\gamma_{HO_2} = 0.070 \pm 0.035$ (ranging from 0.002
to 0.15 across the campaign) was significantly lower than the fixed value of $\gamma_{HO_2} = 0.2$
commonly used in modelling studies. This calculated value was similar, however, to values
calculated for the Wangdu 2014 summer campaign in China (Tan et al., 2020; Song et al.,
2020). Using the calculated $\gamma_{HO_2}$, the OH, $HO_2$ and $RO_2$ radical concentrations were modelled
using the Master Chemical Mechanism, and compared to the measured campaign values, with
and without the addition of $HO_2$ aerosol uptake. Due to the low calculated value of $\gamma_{HO_2}$, and
the high levels of NO, rate of destruction analysis showed the dominant $HO_2$ loss pathway to
be $HO_2 + NO$ for all NO mixing ratios with $HO_2$ uptake not contributing significantly to the
loss of $HO_2$ (< 2 %). However, at the lowest NO mixing ratios (i.e. < 0.1 ppb) $HO_2$ loss onto
aerosols contributed up to a maximum of 29 % of the total $HO_2$ loss. Using the modelled $HO_2$
and $RO_2$ radical concentrations for model scenarios with and without $HO_2$ uptake, showed that
on average the $O_3$ production regime was VOC-limited across the entire campaign with the
exception of several days with low NO mixing ratio where the regime tended towards $NO_x$-
limited, meaning that small changes in $NO_x$ would not have a large effect on the $O_3$ production
for this summer period in Beijing, however changes in $HO_2$ uptake could. While the addition
of the calculated uptake coefficient did not change the overall $O_3$ regime across the campaign,
with the $O_3$ production regime remaining strongly VOC-limited, the transition from a VOC-
sensitive to $NO_x$-sensitive $O_3$ regime occurs at higher $NO_x$. This means that for Beijing, where
the $O_3$ production regime is strongly VOC-sensitive and $NO_x$ levels are high, any policy
looking to reduce $O_3$ via the reduction of $NO_x$ needs to consider concurrent PM reduction
policies which may affect $HO_2$ uptake. In cleaner environments, where $NO_x$ levels are lower,



but aerosol surface area is still high, lower values of $\gamma_{HO_2}$, i.e. less than 0.2, could have a more
significant effect on both overall $HO_2$ concentration and the $O_3$ production regime.
*Data availability.* Data presented in this study can be obtained from authors upon request
([d.e.heard@leeds.ac.uk](mailto:d.e.heard@leeds.ac.uk))
*Author contributions.* LKW, EJS, RWM, CY and DEH carried out the radical measurements.
LKW and EJS developed the model and JED performed the calculations. JDL, FS, JRH, RED,
MS, JFH, ACL, AM, SDW, AB, TJB, HC, BO, CJP, CNH, RLJ, LRC, LJK, WJFA, WJB, SS,
JX, TV, ZS, RMH, SK, SG, YS, WX, SY, LW, PF and XW provided logistical support and
supporting data to constrain the model. JED prepared the manuscript with contributions from
all co-authors.
*Competing interests.* The authors declare that they have no conflict of interest.
*Acknowledgements.* We are grateful to the Natural Environmental Research Council for
funding a SPHERES PhD studentship (Joanna E. Dyson). We are grateful to Tuan Vu for
providing supervision and supporting data.

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
