# Peer review of "production during summertime in Beijing"

_Atmospheric Chemistry and Physics, 2022_

## Author Comment (AC1)

**Impact of HO$_2$ aerosol uptake on radical levels and O$_3$ production during summertime in Beijing**

**Dyson et al.**

**Reply to Anonymous referee #1**

We thank the referee for their comments, addressed individually below. The referee's comments are given in normal type with our response in italics. Changes to the manuscript are then given in red type.

1. It is unclear why the authors examine the HO2 loss pathways not the HOx loss pathways. In Figure 8 and Tables 4&5, HO2+NO is only a radical propagation channel, and does not lead to loss of radicals. So is HO2+O3. HO2+NO is the fast cycling between OH and HO2, and of course they are much faster than other pathways in Figure 8. It seems that the authors should compare radical sinks (peroxide, nitrogen and aerosol uptake) as they did in Section 3.3.3 for O3 sensitivity, as it makes little sense to compare radical propagation channels to radical sink channels.

   *We chose to look at loss pathways specifically, to better understand the effect of HO2 uptake on HO2 concentrations, as is the focus of this study. There was not a large change in absolute HO2 concentration when uptake was included in the model because the uptake channel is not competitive with the propagation reaction, as shown in Figure 7.*

2. The aerosol uptake of HO2. It is unclear why the authors only focus on copper here. In previous studies, it was clear that Cu, Fe and potentially other metals can all contribute to HO2 aerosol uptake, which could make the gamma a lot higher. Was Fe measured in this study? If so, it should be mentioned in Tables 2 and 3. The parameterization used in this study in Equation (1) only includes copper, but it does not necessarily reflect what is happening in the atmosphere. The choices of Equation (1) and fixed value (0.2) seems inadequate to address the role of HO2 aerosol uptake. Given the dataset the authors provided, it would be useful if the authors can provide some observational evidence on gamma(HO2), maybe a plot of obs/mod HO2 as a function of aerosol surface area?

   *As stated by the referee the parameterisation introduced by Song et al., (2021) only includes copper. As such, while aerosol soluble Fe (II) concentration measurements are available for the AIRPRO Summer campaign, and Fe (II) is known to catalyse HO2 loss within the aerosol (Mao et al., 2013), typical values are not included in Tables 2 or 3 as we wished to focus on terms in the Song parameterisation. To our knowledge there is no current parameterisation of the effect of Fe (II) concentrations on the uptake of HO2.*

   *A plot of measured/modelled HO2 as a function of measured aerosol surface area for both MCM_SA and MCM_gamma has been added to the manuscript as SI Figure 4.*

*Given that the parameterisation isn't giving a significant uptake value with just Copper, values ranging from 0.002 to 0.15, we would not expect the addition of Fe to have a large additional effect averaged over the entire campaign.*

*Mao, J., Fan, S., Jacob, D.J. and Travis, K.R., 2013. Radical loss in the atmosphere from Cu-Fe redox coupling in aerosols. Atmospheric Chemistry and Physics, 13(2), pp.509-519.*

3. In Figure 2, was the surface area for dry aerosols or wet aerosols? If it was for dry aerosols, the surface area should be corrected for hygroscopic growth and please provide details.

*The SMPS was run without a dryer, and as such, the aerosol surface area quoted is representative of the real ambient size distribution. We have revised the manuscript to make this clearer:*

*Ln 369: "Online particle sizers were run without a drying inlet to ensure aerosol measurements were as close to real ambient size distributions as possible, and therefore correction for hygroscopic growth was not necessary."*

4. It seems that $RO_2$ uptake was discussed in Section 3.3.3, but $RO_2$ uptake was never mentioned in Section 3.3.1 and Section 3.3.2.

*To our knowledge there are no current lab measurements of $RO_2$ uptake. Uptake of isoprene-$RO_2$ onto ambient aerosols have been measured in Japan (Li et al., 2020). Whilst this could be an important loss of $RO_2$ especially in polluted regions, and for multifunctional $RO_2$ species, there is an uncertainty around this because of that.*

*Li, J., Kohno, N., Sakamoto, Y., Pham, H.G., Murano, K., Sato, K., Nakayama, T. and Kajii, Y., 2022. Potential factors contributing to ozone production in AQUAS–Kyoto campaign in summer 2020: Natural source-related missing OH reactivity and heterogeneous $HO_2/RO_2$ loss. Environmental Science & Technology, 56(18), pp.12926-12936.*

5. Figures 7 and 9 seem redundant

*We have revised the manuscript to move Figures 7 and 9 to the SI.*

6. L305: Henry's law constant for $HO_2$ is temperature-dependent. Is that taken into account here?

*Yes. Henry's law constant was calculated using the average daily temperature measured at the site during the campaign.*

7. Line 570-615 is largely from Sakamoto et al. paper.

*We chose to include this section, while referencing Sakamoto et al, for clarity. We have reduced this section with respect to the original reference.*

---

## Author Comment (AC2)

**Impact of $HO_2$ aerosol uptake on radical levels and $O_3$ production during summertime in Beijing**

**Dyson et al.**

**Reply to Anonymous referee #2**

We thank the referee for their comments, addressed individually below. The referee's comments are given in normal type with our response in italics. Changes to the manuscript are then given in red type.

1. Since the measured and modeled OH, HO2, and RO2 concentrations have been discussed in detail in Whalley et al., 2021, the authors should focus this paper on the impact of HO2 uptake on the modeled concentrations. In that light, I would recommend removing Section 2.2 and referencing the Whalley et al., 2021 ACP paper (and updating the reference to the discussion paper).

   *We have revised the manuscript to move Section 2.2 to the SI.*

2. I would also suggest moving the description of the LN/Q and absolute O3 sensitivity calculation (lines 570-603) to section 2 after the model description, and instead focusing on the results in Section 3.

   *We have revised the manuscript, moving the description of LN/Q and O3 sensitivity to section 2.*

3. The authors provide a brief description regarding potential reasons for the discrepancy between the modeled radical concentrations with the measurements, which are discussed in detail in Whalley et al. (2021). However, at first read the description here does not appear to be consistent with the description in Whalley et al. For example, line 462 states that the overprediction of HO2 by the model may be due to "an under-prediction in the rate of reaction of RO2 with NO to produce a different RO2 species…" while the conclusion in Whalley et al. 2021 is that the "propagation rate of RO2 to HO2 may be substantially slower than assumed." While I believe the reasoning is consistent between the two papers, the wording here could be clarified to remove any potential confusion.

   *We agree with the referee and for clarity have revised line 460 onwards in the manuscript to read:*

   *"the over-prediction of $HO_2$ could be due, in part, to the propagation rate of $RO_2$ to $HO_2$ being significantly slower than currently included in the model. This could be due to a lack of understanding of the rate of reaction of $RO_2$ with NO to produce different $RO_2$ species, i.e. $RO_2 + NO \rightarrow RO_2$', which would lead to propagation of $RO_2$ to different, more oxidised $RO_2$ species, competing with the recycling of $RO_2$ via $RO_2$ to give $HO_2$. It is also possible, that the overestimation in the propagation rate of $RO_2$ to $HO_2$ could be due to a lack of $RO_2$ autoxidation pathways included within the model which could lead to the formation of highly oxygenated molecules as opposed to $HO_2$. The higher, measured $RO_2$ concentrations could, therefore, suggest that the lifetime of total RO2 is longer than currently considered within the model."*

4. Similar to that described in Sakamoto et al. (2019), the authors include uptake of RO2 radicals into account when analyzing the impact of aerosol uptake on ozone production sensitivity. Given the authors suggestion that the overprediction of HO2 and underprediction of RO2 is due to isomerization of complex RO2 or RO radicals that effectively increases the lifetime of RO2 radicals and slows the propagation of RO2 to HO2, can the authors comment on whether uptake of RO2 radicals in this scenario could impact the concentration of RO2 radicals and the rate of ozone production? What effective lifetime of RO2 radicals would heterogeneous uptake be competitive and impact RO2 concentrations? Perhaps include a plot similar to Figure 8 for RO2 loss to address this?

*We would expect that if the uptake of $RO_2$ to aerosol is significant, then the concentration of $RO_2$ radicals within the model would decrease, in turn decreasing the $HO_2$ concentration due to an increase in $RO_2$ radical sinks. This would slow the propagation of $RO_2$ to $HO_2$, and therefore decrease the $HO_2$ radical overprediction within the model if this process is included. Looking at Table 4 in this work, at maximum 7.3 % of total loss of HO2 in MCM_gamma is via uptake. Li et al., (2020) measured ambient uptake of isoprene-RO2 and gave the mean loss of HO2 onto aerosols as 0.0014 $s^{-1}$, just under double the loss rate of RO2 onto aerosols, at 0.0008 $s^{-1}$. Looking at Figure 3 in Whalley et al., (2021), a full comparison of median production and destruction rates for OH, HO2, total RO2 and ROx for the Beijing Summer campaign is given, showing the gas phase loss of RO2 to be much higher than HO2. This, coupled with the lower loss rate of RO2 measured by Li et al., (2020), suggests the RO2 uptake would not be significant to impact RO2 concentrations for this campaign.*

*Li, J., Kohno, N., Sakamoto, Y., Pham, H.G., Murano, K., Sato, K., Nakayama, T. and Kajii, Y., 2022. Potential factors contributing to ozone production in AQUAS–Kyoto campaign in summer 2020: Natural source-related missing OH reactivity and heterogeneous HO2/RO2 loss. Environmental Science & Technology, 56(18), pp.12926-12936.*

*Whalley, L.K., Slater, E.J., Woodward-Massey, R., Ye, C., Lee, J.D., Squires, F., Hopkins, J.R., Dunmore, R.E., Shaw, M., Hamilton, J.F. and Lewis, A.C., 2021. Evaluating the sensitivity of radical chemistry and ozone formation to ambient VOCs and NO x in Beijing. Atmospheric Chemistry and Physics, 21(3), pp.2125-2147.*